# Engineered dityrosine-bonding of the RSV prefusion F protein imparts stability and potency advantages

Sonal V. Gidwani[1], Devarshi Brahmbhatt[1], Aaron Zomback[1], Mamie Bassie[1], Jennifer Martinez[1], Jian Zhuang[1,5], John Schulze[2], Jason S. McLellan [3], Roberto Mariani[1,6], Peter Alff[1], Daniela Frasca[4], Bonnie B. Blomberg[4], Christopher P. Marshall[1] & Mark A. Yondola [1] ✉

Viral fusion proteins facilitate cellular infection by fusing viral and cellular membranes, which involves dramatic transitions from their pre- to postfusion conformations. These proteins are among the most protective viral immunogens, but they are metastable which often makes them intractable as subunit vaccine targets. Adapting a natural enzymatic reaction, we harness the structural rigidity that targeted dityrosine crosslinks impart to covalently stabilize fusion proteins in their native conformations. We show that the prefusion conformation of respiratory syncytial virus fusion protein can be stabilized with two engineered dityrosine crosslinks (DT-preF), markedly improving its stability and shelf-life. Furthermore, it has 11X greater potency as compared with the DS-Cav1 stabilized prefusion F protein in immunogenicity studies and overcomes immunosenescence in mice with simply a high-dose formulation on alum.

Dityrosine bonds naturally form to provide structural rigidity in proteins such as elastin, collagen, and resilin. They are also found in bamboo, the joints of grasshoppers and dragonflies, and the human aorta[1–5]. Additionally, they have been engineered into proteins such as silk fibroin to improve visco-elastic properties and resilience[6]. Dityrosine bonds are stable even under extreme conditions, including acid hydrolysis and boiling in reducing sample buffer[7,8]. Their formation can be catalyzed by several different mechanisms including enzymatic (peroxidase) processes[7–13]. The enzymatic dityrosine crosslinking reaction propagates through a resonance-stabilized free radical mechanism. The reaction only forms crosslinks between tyrosine sidechains, and because dityrosine bonds are zero-length they can only form between tyrosine residues in structural proximity. Therefore, this reaction can be used to identify protein: protein interactions (e.g., in capsid studies of the adeno-associated virus)[14]. We have leveraged the specificities of this reaction, and developed a technology to engineer targeted dityrosine crosslinks at specific positions in a protein structure in order to stabilize proteins and lock desirable protein conformations[15–17]. Unlike disulfide bonds, dityrosine bonds are introduced through an enzymatic step in manufacturing by which the protein is stabilized after it is fully folded, minimizing misfolding/ aggregation that often occurs with engineered disulfide bonds[18].

Many human pathogenic viruses (e.g., influenza, coronaviruses, Ebola, HIV, and the respiratory syncytial virus−RSV) are enveloped, meaning they have an outer lipid bilayer derived from the host cell. Therefore, fusion of viral and cellular membranes is a key step in the entry of all enveloped viruses and is accomplished by virally encoded, fusion proteins that fuse lipid bilayers by transitioning from their prefusion to their postfusion conformations. In recent years it has become clear that fusion proteins elicit antibody (Ab) responses that neutralize viral infection, a powerful and predictive measure of vaccine efficacy. Fusion proteins in their prefusion conformation expose

[1]Calder Biosciences Inc., Brooklyn Army Terminal, Brooklyn, NY, USA. [2]Molecular Structure Facility, University of California, Davis, Davis, CA, USA. [3]Department of Molecular Biosciences, University of Texas at Austin, College of Natural Sciences, Austin, TX, USA. [4]Department of Microbiology and Immunology, University of Miami, Miami, FL, USA. [5]Present address: Department of Medicine, Donald and Barbara Zucker School of Medicine at Hofstra/ Northwell, Manhasset, NY, USA. [6]Present address: CUNY Kingsborough Community College, Brooklyn, NY, USA. ✉e-mail: yondola@calderbiosciences.com

potently neutralizing epitopes to the immune system and these prefusion-specific epitopes give rise to the most potently neutralizing and protective Ab responses, likely because only fusion proteins in their prefusion conformation on the surface of infecting virus particles mediate viral entry. Antibodies that bind the prefusion conformation with high avidity and affinity block the fusion protein's function, and thus prevent infection. However, one of the major challenges to developing recombinant, soluble prefusion vaccine immunogens is that the prefusion conformation is spring-loaded and metastable, and often readily transitions irreversibly into the postfusion conformation ("postF").

Structure-based design has yielded proteins that provide proof of principle that prefusion vaccine immunogens are, indeed, the most potent (e.g., RSV DS-Cav1)[19]. However, these designs are often only partially stabilized, and in addition to prolonging shelf-life and reducing cold-chain requirements, we believe that further improving thermostability also prolongs in vivo exposure of prefusion-specific epitopes, strengthens the antigenic signal of these epitopes, and thus enhances B-cell responses and affinity maturation. Thus, fusion proteins better stabilized in the prefusion conformation should yield higher-affinity and higher-titer Ab responses to prefusion-specific epitopes and provide better protection. Dityrosine ("DT") crosslinking provides a means to stabilize fusion proteins in their prefusion conformation that goes beyond the limits of traditional, mutation-based designs (for example, disulfide engineering, and cavity filling mutations)[18,19]. By focusing Ab responses on prefusion-specific epitopes, dityrosine-stabilized immunogens can elicit more potently neutralizing Ab responses, resulting in better vaccine immunogens.

Respiratory Syncytial Virus (RSV) infects humans repeatedly throughout life. Newborn children, the elderly, and immune compromised patients are particularly vulnerable to more severe disease[20–22]. In newborn children, RSV infection often results in protracted and enhanced respiratory disease, which can result in prolonged respiratory difficulty throughout childhood and adolescence. However, development of an RSV vaccine was hindered by an early clinical trial in which children vaccinated with a formalin-inactivated vaccine (FI-RSV) experienced an exaggerated immune response to subsequent RSV infection—i.e., vaccine enhanced disease (VED)[23]. VED was characterized by elevated eosinophilic responses to infection, airway hyperreactivity, and excessive mucus production[23,24]. These potentially fatal responses were caused by poorly neutralizing Ab responses with immune complex deposition in the lungs and Th2-type cytokine responses to vaccination in RSV naïve children[24,25]. Eliciting high titers of prefusion-specific, neutralizing antibodies is still considered the best vaccine strategy to protect the elderly and infants via maternal-to-infant vaccination, whereby the importance of T cells to the protection of the elderly should not be discounted[26–30].

The prefusion conformation of the RSV F (fusion) protein is among the most labile of the fusion proteins. However, an RSV prefusion F (preF) subunit vaccine has been demonstrated to be one of the most promising approaches, since it elicits highly potent neutralizing antibody responses, and because its sequence is highly conserved between strains. In fact, the first two RSV vaccines to obtain licensure are protein subunit vaccines[31–33]. Significant progress has been made stabilizing RSV preF, starting with the design and characterization of the Vaccine Research Center's (VRC/NIH) DS-Cav1 molecule[19]. DS-Cav1 elicits substantially higher neutralizing antibody titers than the RSV F protein in its postfusion conformation[19,28,34]. Several 2nd-generation RSV subunit vaccines are being developed that further improve upon DS-Cav1's stability[35–42]. Current RSV prefusion stabilized first generation vaccines have built upon the success of DS-Cav1 and have now achieved full licensure for the elderly and in maternal vaccination[31–33,43]. These exciting first generation molecules have obtained high overall efficacy and have also substantially de-risked the clinical development pathway for subunit RSV prefusion vaccines.

Nevertheless, ample room for improvement exists. Pfizer's maternal vaccine does not substantially outperform the protection afforded by monoclonal antibody administration in infants; and GSK's leading vaccine for the elderly does not adequately protect in the 80+ elderly or the frail where they achieve only 34 and 14% efficacy, respectively and lack statistical significance. These immunosenescent populations are precisely the most likely to be hospitalized and have poor clinical outcomes and vaccines therefore need to elicit potent responses in these groups. This remains an important unmet medical need. Using influenza vaccines formulated for the elderly as an example, surmounting immunosenescence is achievable by modulation of the dose and/or adjuvant[44,45]. Applying these approaches to the most potent immunogen has the greatest chance of success. Herein we report the application of targeted dityrosine crosslinking to improve the efficacy of a fusion protein-based subunit vaccine. We have harnessed the structural rigidity that dityrosine (DT) bonds can impart to generate an RSV preF subunit vaccine that more stably holds its prefusion conformation.

Our prefusion F molecule, DT-preF, comprises two, targeted dityrosine crosslinks that lock the preF conformation to elicit Ab responses focused on neutralizing epitopes[29,46]. Since dityrosine bonds only form between tyrosine sidechains in close proximity, we have engineered our DT-preF molecule through structure-based design and site-directed mutagenesis to stabilize two of the most highly prefusion-specific epitopes (Site Ø, and the Site IV/V interface). Therefore, the dityrosine bonds formed in DT-preF are not present in the DS-Cav1 molecule. The previously described DS-2 molecule targeted a disulfide bond to the Site IV/V interface and improved upon the potency of DS-Cav1 approximately 4-fold but expresses poorly and has not progressed into clinical development as a subunit vaccine. Dityrosine induced thermostability stabilizes the prefusion conformation of DT-preF both in vitro and in vivo. Our data demonstrate that DT-preF elicits 11X higher neutralizing Ab titers than those elicited by DS-Cav1, and can overcome immunosenescence in a mouse model using a high dose and the well tolerated alum adjuvant. DT-preF is also highly potent in cotton rats and elicits high neutralizing antibody titers against RSV A and B strains and achieves sterile lungs upon challenge with the WT RSV Long strain. These preclinical data suggest that dityrosine-stabilized DT-preF has clear advantages that will fill the remaining gaps in the protection afforded by first generation RSV vaccines.

## Results

### Immunogen design based on dityrosine crosslinking

RSV F protein is expressed as $F_0$, a single polypeptide with two furin cleavage sites. The RSV F protein has between 5 and 6 N-linked glycosylation sites on the precursor molecule[47]. Proteolytic maturation of the F trimer removes 27 amino acids, yielding the $F_1$ and $F_2$ cleavage products, which together form a protomer of the F trimer. The mature protomer contains 3 N-linked glycans (2 on the $F_2$ subunit and 1 on the $F_1$ subunit)[48]. We engineered Tyrosine (Tyr) substitutions in the soluble F glycoprotein to preserve protective epitopes recognized by prefusion-specific neutralizing Abs[19]. We targeted these substitutions based on the crystal structure of the RSV F protein complexed with the prefusion-specific D25 monoclonal antibody (mAb), and the DT bonding designs were screened in the previously described Cav1 background in order to build on a framework sufficiently stable for analysis (RSV F A2)[19]. Soluble forms of these variants were designed for intra and/or intermolecular crosslinking and expressed by transient transfection in HEK293T cells and screened for expression and antigenicity by ELISA (Fig. 1A, B). Motavizumab, a mAb that has a high affinity for both pre- and postfusion F, was used to measure total protein expression. Prefusion antigenicity was measured using mAbs D25, AM14, and MPE8 to map sites Ø, the IV/V interface, and site III binding, repectively (Fig. 1B). A few selected dityrosine mutations that were localized to critical epitopes for neutralization and

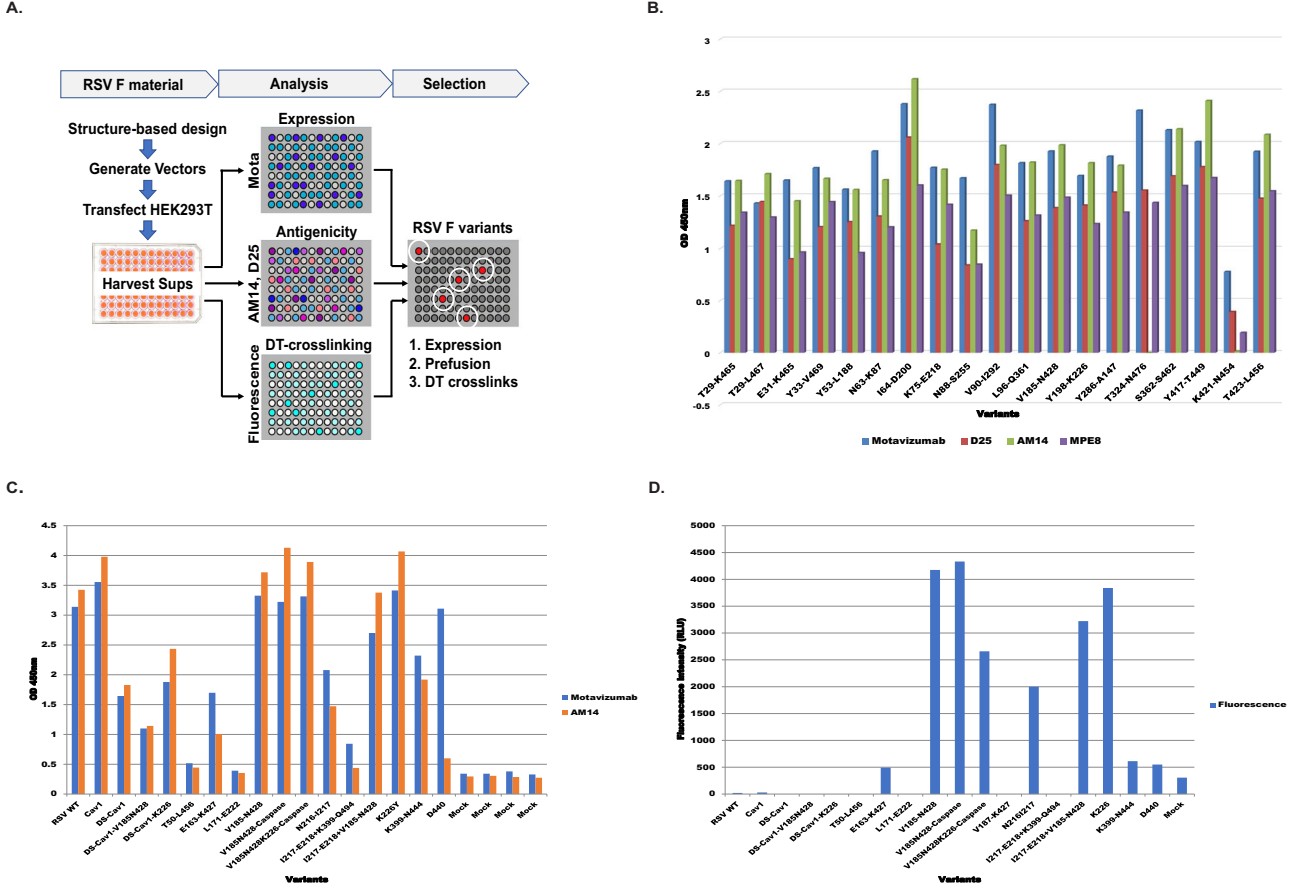

**Fig. 1 | Screening of dityrosine pairs within the RSV F protein for expression, conformation, and dityrosine bond formation. A** RSV F DT-mutants were transfected into 293T cells in a 96-well plate format, harvested on day 3, and supernatants were used in ELISA based total protein (Motavizumab), and conformational (AM14, D25, MPE8) analysis. Crosslinking analysis by fluorescence intensity measurement after crosslinking of proteins in supernatants was also included in the screen. **B** Graphs of ELISA data showing a representative subset of the initial screening of expression level and conformation using Motavizumab (Site II), D25 (Site Ø), AM14 (Site IV/V) and MPE8 (Site III) respectively. Source data from the graphed, representative experiment are provided as a Source Data file. This experiment was performed twice. **C** Graphs of ELISA data of second-tier screening for expression using Motavizumab and preF antigenicity using AM14 done in conjunction with (**D**). Source data are provided as a Source Data file, this experiment was performed in duplicate. **D** Fluorescence intensity (ex320nm/em405nm) measurements were taken before and after crosslinking of potential hits using supernatants from transfected 293T cells, where protein signal was normalized to the DS-Cav1-V185N428 variant (demonstrating low expression). This graph indicates the mutant-specific signal post background subtraction of mock-transfected samples. Source data are provided as a Source Data file, this experiment was performed in duplicate.

demonstrated good levels of antigenicity were also tested in the DS-Cav1 background (Fig. 1C). For further characterization, the top 14 variants and WT/Cav1/DS-Cav1 controls were screened for antigenicity and fluorescence in supernatants of transfected 293T cells (Fig. 1C and 1D). Since dityrosine bonds are fluorescent with unique excitation and emission maxima (ex320/em405), screening by this method enables the detection of both intra and inter-protomeric dityrosine bonds. After the crosslinking reaction, DT bonds were detected by fluorimetry (Fig. 1D). As shown in Fig. 1D, using a representation of samples normalized for protein expression, several hits were identified in the fluorescence intensity screen. While many of the designed constructs demonstrated significant binding to conformation-specific antibodies in the absence of crosslinking, we prioritized crosslinks that would preserve epitopes that elicit the most potently neutralizing prefusion-specific antibodies, namely Site Ø-stabilized by the K226Y mutation pairing with endogenous Y198, and the AM14 binding site (IV/V) interface. Given the utility of AM14 as the most prefusion-specific mAb known for RSV F, we targeted several crosslinks precisely in the AM14 binding site but sought to maintain some level of AM14 binding post-crosslinking[46]. The variant designed with tyrosine substitutions at Valine 185, Lysine 226, and Asparagine 428, termed preF^C (Fig. 2A, bottom right), which is a descendent of Cav1 (Fig. 2A, top), and related

to DS-Cav1 (Fig. 2A, bottom left) through the shared use of the Cav1 mutations, possessed near WT expression levels and a favorable AM14 binding profile before and after crosslinking (Fig. 2C, D). This molecule consists of two engineered dityrosine bonds: V185Y pairing with N428Y, and K226Y pairing with the endogenous tyrosine at Y198 (Fig. 2A). These crosslinks stabilize the antigenic site IV/V interface (AM14 binding region) and prefusion-specific Site Ø (5C4 binding region), respectively (Fig. 2A, B)[28,46,49]. preF^C has three additional point mutations that targeted the Foldon region designed to further enhance stability- these are L512V, L513V, and Y519F. We further characterized DT-preF binding properties with purified proteins as compared to DS-Cav1 using antibodies that map to key epitopes including Site Ø (D25), Site II (Motavizumab), and Site III (MPE8) (Supplementary Fig. 1). Additionally, we characterized the fluorescence properties of purified DT-preF relative to protein concentration under native conditions and can demonstrate linearity in the range of 19–152 μg/mL (Supplementary Fig. 1).

**Dityrosine crosslinking is highly specific for engineered sites**
To map the location of the observed dityrosine bond, we submitted the protein for LC-MS-MS. To simplify the analysis, purified crosslinked and uncrosslinked samples were submitted for comparison. Unique

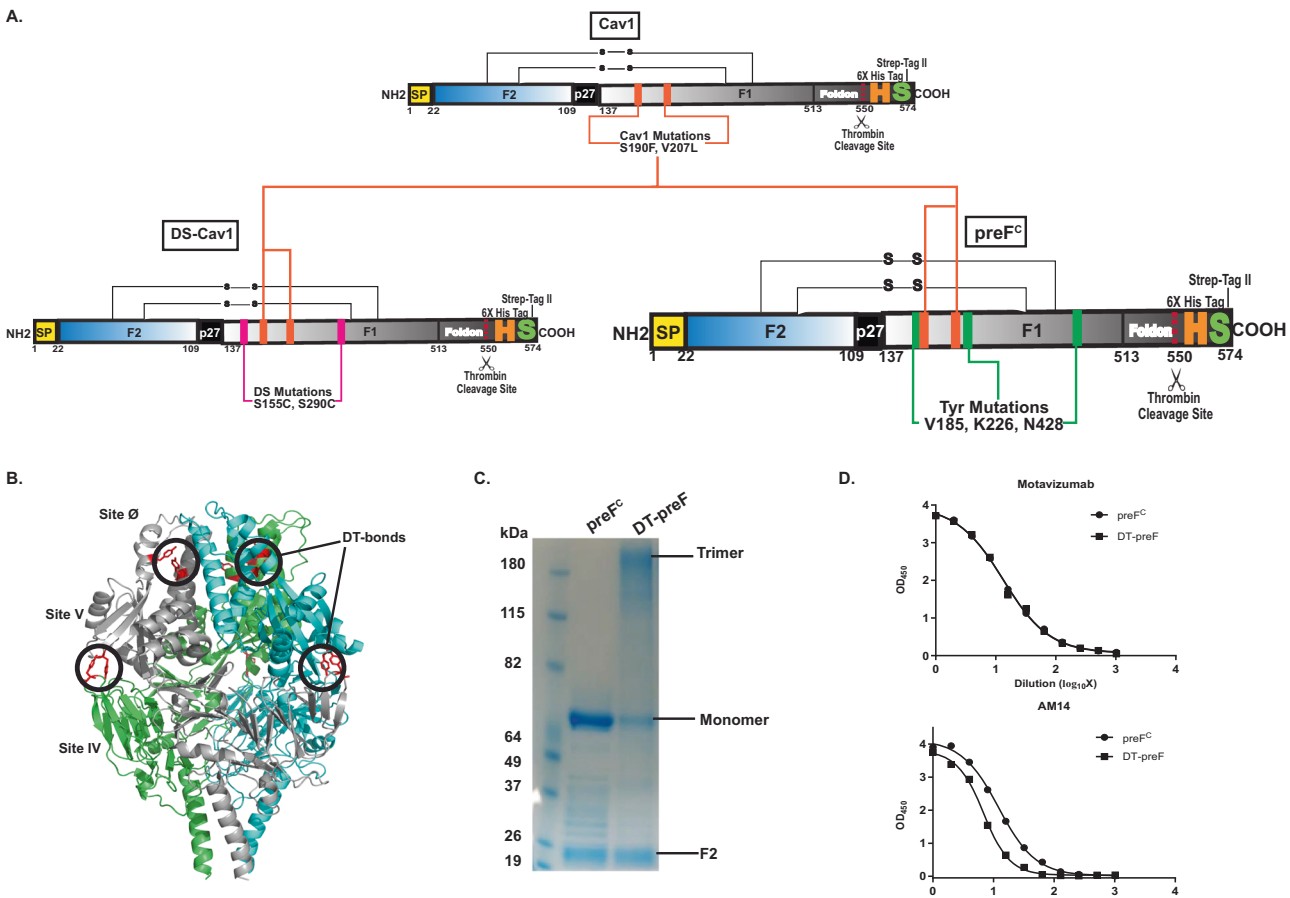

**Fig. 2 | Design and characterization of dityrosine crosslinked DT-preF molecule. A** TOP: RSV F parent molecule (Cav1) with Cav1 mutations indicated; Bottom Left: DS-Cav1 successor of Cav1 parent with additional DS mutations indicated. Bottom Right: preF$^C$ with tyrosine mutations indicated for dityrosine pairing. Note: K226Y pairs with endogenous tyrosine Y198. Amino acid numbering is indicated below the stick diagram. The "H", "S", and scissors represent 6X His tag, Strep-II tag, and Thrombin cleavage site, respectively. **B** Crystal structure of DS-Cav1 protein with preF$^C$ mutations modeled on the structure. DT bond locations are circled indicating the intramolecular crosslink preserving site ∅ (5C4, D25) and the intermolecular crosslink preserving the site IV/V interface (AM14). **C** Coomassie Blue Protein staining of SDS-PAGE separated preF$^C$ and DTpre-F under reducing conditions before and after crosslinking. Monomer, trimer, and F2 are indicated. Similar gel shifts have been observed consistently (>100 independent experiments). **D** ELISA binding curves of preF$^C$ and DT-preF using primary antibodies Motavizumab and AM14 before and after crosslinking. Source data are provided as a Source Data file, the average data from 2 different crosslinking experiments is graphed.

peptides were identified in the crosslinked sample, and these were selected and sequenced by tandem MS. The intermolecular crosslink V185Y-N428Y was identified in the mass spectrum following comparative peptide fragment analysis and the sequence of crosslinked peptides was confirmed by tandem mass spectrometry (Fig. 3A–C). The intramolecular bond was not able to be identified in the mass spectrometry analysis. We characterized the protein further by amino acid analysis and analytical SEC under denaturing conditions. Amino acid analysis confirmed dityrosine's presence in the sample and between 4 and 5 dityrosine bonds were present on average per molecule (Fig. 4A). Since only 3 intermolecular bonds can form per trimer, these data suggest that additional dityrosine bonds form in the crosslinked protein. We further characterized our crosslinked molecule by analytical size exclusion chromatography (SEC) under denaturing conditions with in-line fluorimetry and compared it with uncrosslinked preF$^C$ (Fig. 4B, left bottom and top, respectively). Separation of the monomeric and multimeric peaks under denaturing conditions and observation of the fluorescence associated with both peaks demonstrates that an intramolecular bond is forming in the crosslinked molecule that is not observable by mass spectrometry (Fig. 4B, right top and bottom). In support of this finding, we subjected both DS-Cav1 and the Cav1 control molecules to the crosslinking reaction conditions. As shown in Supplementary Fig. 2A, only the preF$^C$

exposed to these conditions resulted in intermolecular bond formation as apparent by the MW shifts observed under denaturing conditions. The Cav1 protein represents the closest control to our molecule since preF$^C$ only differs from Cav1 by 3 mutations to tyrosine to allow crosslink formation and the 3 stabilizing point mutations near the Foldon domain. Therefore, we confirmed the lack of significant MW shifting on crosslinked and purified CAV1 protein using SDS-PAGE with Coomassie staining and western blot analysis (Supplementary Fig. 2B). Since intramolecular bonds would not be apparent by gel shift, we further subjected the crosslinked and purified Cav1 protein to denaturing HPLC analysis and can demonstrate that fluorescence is absent in the monomeric peak as compared with a similarly crosslinked and purified DT-preF protein subjected to the crosslinking reaction (Supplementary Fig. 2B, C).

## Dityrosine bonding of preF$^C$ confers stability improvements that translate into prolonged shelf life

In order to demonstrate the stability imparted by the DT crosslink, we subjected our DT-preF molecule to differential scanning fluorimetry using the Prometheus Panta instrument. As described in the Supplementary materials and methods section and demonstrated in Supplementary Fig. 3A, the crosslinked molecule exhibited a melting temperature increase of 25 °C as compared with the uncrosslinked

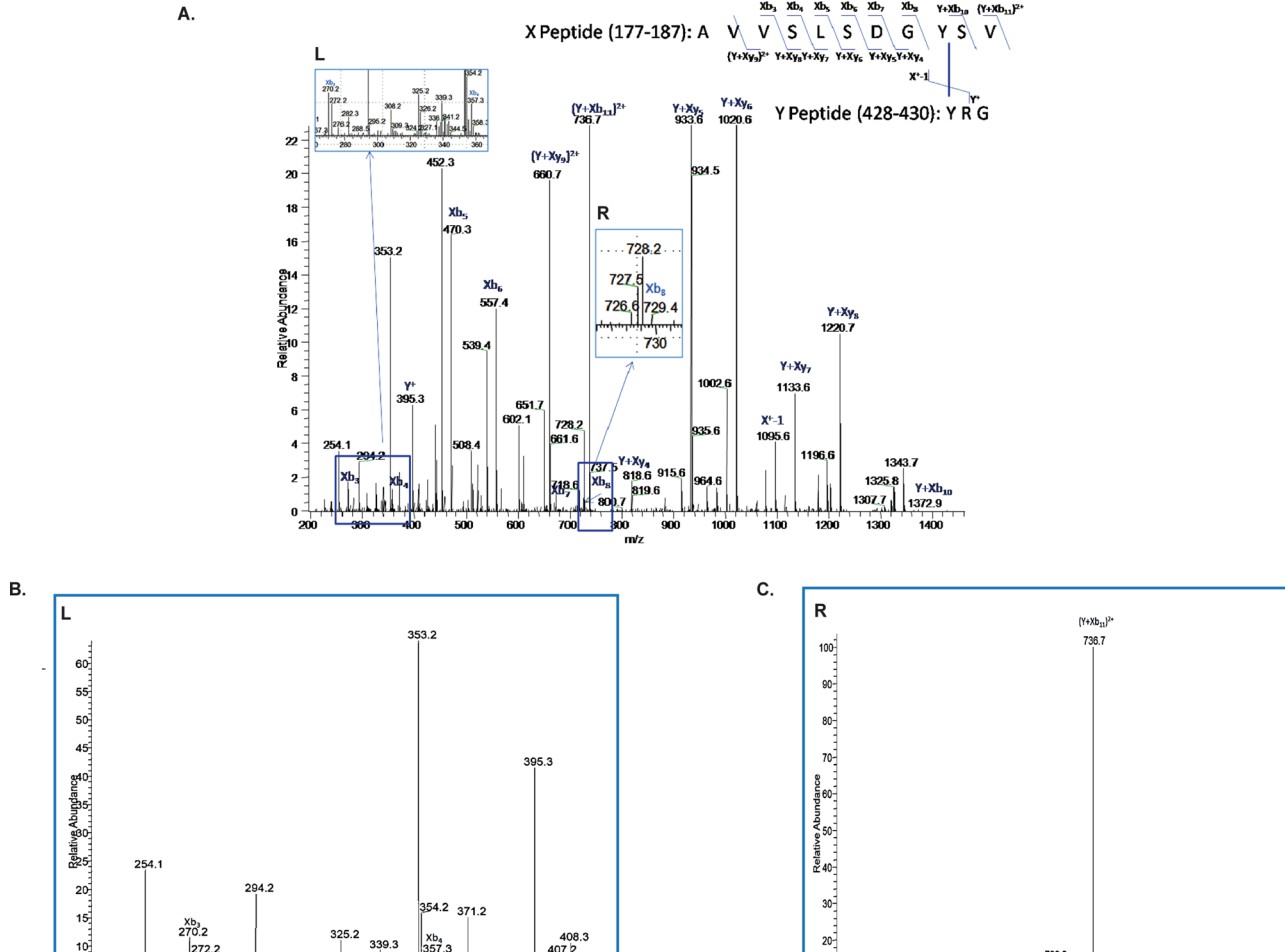

**Fig. 3 | Tyrosine modification analysis of DT-preF using NanoLC-ESI-MS/MS. A** MS/MS spectrum of conjugated peptide indicating a Y-Y linkage (Y185-Y428) between the 'X' peptide (AVVSLSDGYSV 177–187) and 'Y' peptide (YRG 428–430) following digestion of the glycoprotein with trypsin, chymotrypsin, and pepsin. The ion series is indicated at the top right of the figure, while the mass spectrum is beneath with the ion series labeled. **B** Zoomed in view of the Left (L) insert (**C**) Zoomed in view of the Right (R) insert. Mass spectrometry analysis to find cross-linked species was performed a single time due to 98.5% coverage of the protein.

preF$^C$ molecule. We next tested the stability of our DT-preF molecule in vitro at 4 °C in a time-course experiment. We thawed purified DT-preF and DS-Cav1, and normalized protein concentration and buffer composition. Proteins were then incubated at 4 °C for 3 and 5 weeks, and compared to Motavizumab-normalized, freshly thawed (day 0) protein on the day of each ELISA (Fig. 5A). Notably, while DT-preF's binding to AM14 remained stable at weeks 3 and 5, DS-Cav1's binding continued to fall and reached an 82% loss overall by week 5 (Fig. 5A). Based on these results, we expanded this analysis in terms of time frame and epitopes probed to include D25 (Site Ø), and also developed a highly prefusion-specific sandwich ELISA that utilizes both 5C4 (Site Ø) and AM14 (Site IV/V) antibodies to measure the prefusion conformation. As apparent in Supplementary Fig. 3B, DT-preF maintains the prefusion conformation even out to 11 weeks of storage at 4 °C using both ELISA assays. While DS-Cav1 maintains binding to D25 over this period, the 5C4/AM14 sandwich ELISA we have established clearly demonstrates significant loss of the prefusion conformation during 4 °C incubation over the same time period. This result demonstrated marked 4 °C stability improvement for the DT-preF molecule in vitro so we tested the impact of this result in murine immunogenicity experiments. We injected BALB/c mice intramuscularly with freshly thawed DT-preF and DS-Cav1 immunogens and identical immunogens after 4 weeks of cold storage formulated on Advax$^{SM}$ [50–52]. Advax was chosen for this study since at the time we were trying to elicit more Th-balanced responses for their potential safety advantage. Neutralization assays performed with the mouse serum showed a statistically significant loss of potency (57.6%) for the DS-Cav1 protein but not the DT-preF after 4 weeks of incubation at 4 °C (Fig. 5B, C). Hence, dityrosine crosslinking of DT-preF resulted in significantly improved stability and shelf life.

## DT-preF is highly potent in mice
Hypothesizing that improved stability results in better affinity maturation and improved potency, we compared DT-preF and DS-Cav1 formulated with the Th2-skewing adjuvant alum in mice in order to maximize serum antibody responses. Neutralizing Ab titers (NTs, expressed as reciprocal IC$_{50}$'s) of mouse sera demonstrated that DT-preF elicited 11.1-fold higher geometric mean NTs than DS-Cav1 (Fig. 6A) using a traditional prime: boost regimen. It was important to determine whether these enhanced responses were due to a larger overall anti-F antibody response, or whether the DT-preF immunogen elicited a

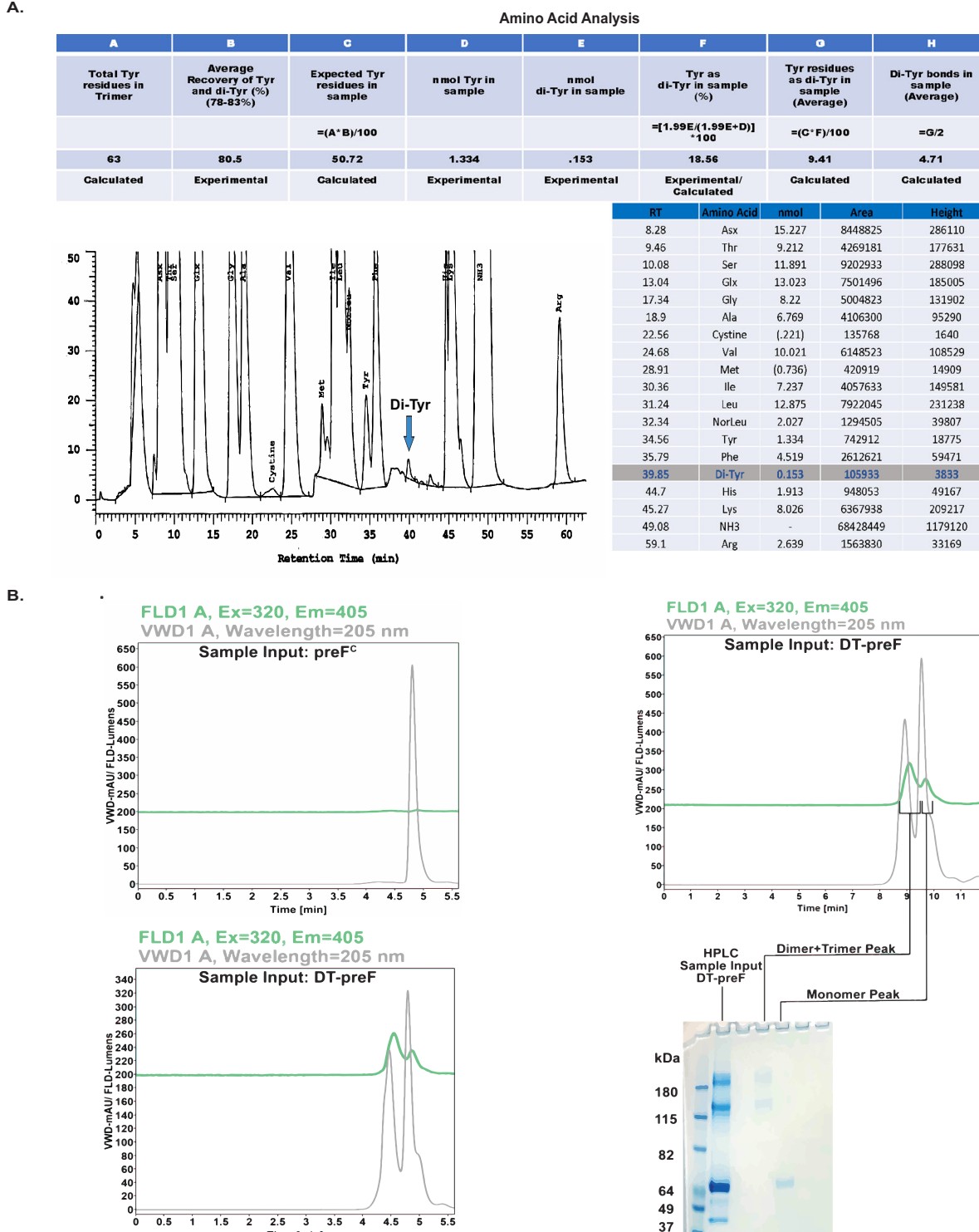

**Fig. 4 | Biochemical/biophysical characterization of the DT-preF molecule.**
**A** Amino acid analysis of DT-preF. Samples were acid-hydrolyzed and analyzed for amino acid content using a Concise AminoSep Beckman Style Na+ column and a Hitachi analytical HPLC. Table (top) with calculated Dityrosine (Di-Tyr) bonds in the DT-preF sample utilizing the experimentally determined percentage recovery and quantification information. Chromatogram (bottom left) indicating the detection of the dityrosine peak in the DT-preF sample. Table (bottom right) indicating list of amino acids detected in the analysis with the respective peak attributes/quantification. Amino acid analysis was performed a single time. **B** Chromatograms of uncrosslinked F protein containing the DT mutations, preF[C] (top left) and DT-preF (bottom left) showing fluorescence (green) and UV (gray) traces at increasing retention times (min) following separation using analytical size exclusion chromatography under reducing and denaturing conditions. Chromatogram showing fluorescence (green) and UV (gray) traces of DT-preF (top right) and its corresponding SDS-PAGE analysis gel (bottom right) indicating separation of the dimer/trimer peak from the monomer peak, as indicated. This separation between multimers and monomers observed on SDS-PAGE analysis gels has been observed and reproduced in >3 independent experiments.

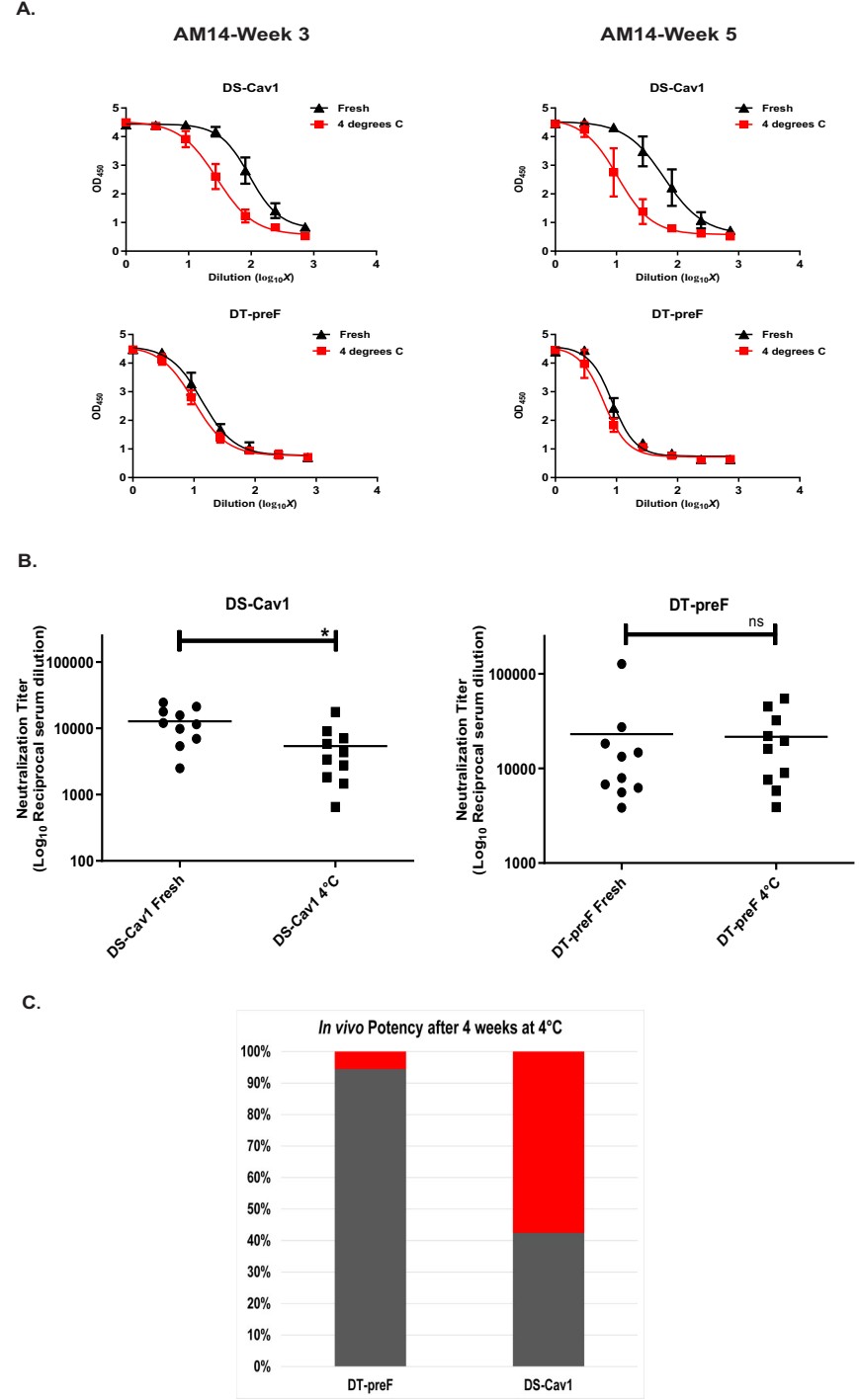

**Fig. 5 | In vitro stability of dityrosine crosslinked DT-preF molecule and its effects on potency in vivo. A** In vitro stability of DS-Cav1 (top) and DT-preF (bottom) incubated at 4 °C vs unincubated (fresh) at the indicated timepoints. Binding curves for AM14 (performed in triplicate) are presented on Motavizumab-normalized proteins for each timepoint. Data are presented as mean values ± SD. ELISAs were run in triplicate using 2 different crosslinked protein preparations. **B** Dot plots of the neutralization titer data as reciprocal of IC50's with mean titers indicated as a bar. Groups of animals were compared on the distribution of the outcome using the Wilcoxon Rank Sum test. Sample size 'n' to derive statistics = 10, 6–8-week-old, CB6F1/J female mice/ group. * Indicates $p = 0.0420$ significance. Neutralization assays were run in quadruplicate. **C** Graphical representation of the data in B with DT-preF and DS-Cav1 groups' potency loss as a percentage of their freshly thawed values (set at 100%) was compared in terms of mean neutralization titers after incubation at 4 °C for 4 weeks. The percentage loss after 4 weeks of each sample is shown in red and the remaining potency is in gray. For (**A**), (**B**), and (**C**) source data are provided as a Source Data file.

better-quality response, with a higher percentage of the total antibodies eliciting more potent viral neutralization. Figure 6B shows that only a 1.7-fold increase in total anti-prefusion F binding titers were detected, demonstrating that DT-preF elicited better-quality responses since a larger percentage of the elicited antibodies neutralize the virus.

## A high-dose DT-preF vaccine formulated on alum overcomes immunosenescence in aged mice

The elderly represent a crucial patient population at risk for severe complications from RSV infection and specifically the 80+ age bracket and/or the frail are at the greatest risk. Immune responses in these

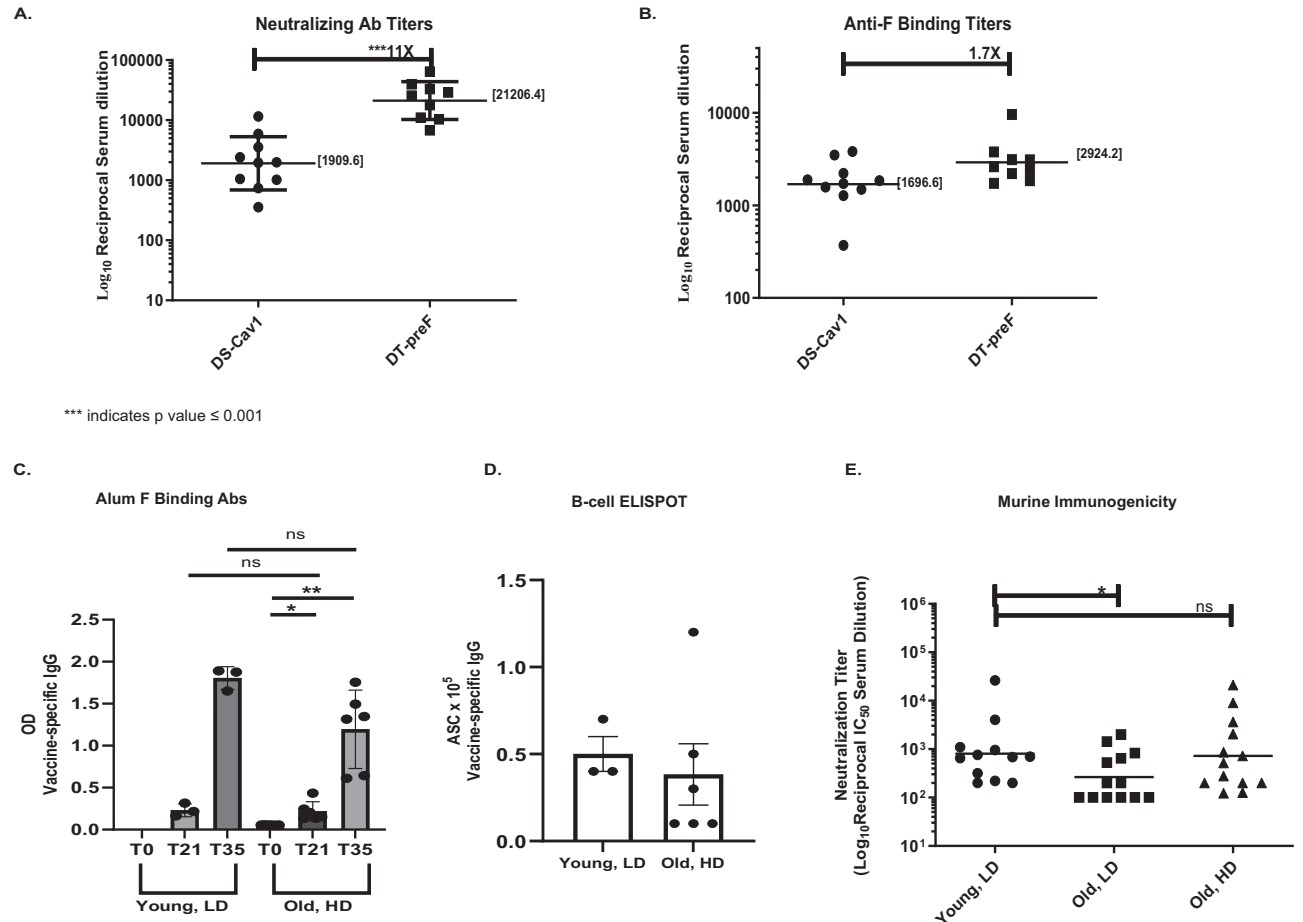

**Fig. 6 | DT-preF is a highly potent immunogen in mice. A** Mouse serum neutralization titers were measured by Renilla Luciferase assay after vaccination with 2.3 μg of the indicated immunogens formulated on Alhydrogel to compare potency of DT-preF and DS-Cav1. The reciprocal of the IC50 neutralizing antibody titers are graphed as dot plots. Geometric mean titers are graphed and the numerical value is specified on the graph as a bar. Error bars represent the geometric SD. Groups of animals were compared on the distribution of the outcome using the Mann-Whitney test with a *p* value of 0.0002. Sample size '*n*' to derive statistics = 10, 15-week-old, female CB6F1/J mice/group; assays were run in duplicate. **B** Anti prefusion F binding titers were measured by ELISA using the serum obtained in (**A**) and prefusion F as the capture antigen. The EC50's were plotted as dot plots with the geometric mean titers graphed, represented as a bar and the numerical value is specified on the graph. Sample size '*n*' to derive statistics = 10,15-week-old, female CB6F1/J/ group; assays were run in duplicate. **C** Young (4 months) and old (17 months) female, BALB/c mice were immunized with our DT-preF immunogen on alum at a low dose (LD; 10 μg) and a high dose (HD; 45 μg), respectively. Anti- preF antibody responses were quantified from serum samples collected on day 0 (prebleed), day 21 (prime only), and day 35 (prime-boost) by ELISA-based assay and plotted as bar graphs with the individual titers indicated by dots. Mean comparisons were performed by a 2-tailed paired (within group) or unpaired (between groups) Student's *t* test using GraphPad Prism software. Error bars represent the SEM. *p* values of 1 and 2 star significances are *p* = 0.0186 and *p* = 0.0019, respectively. **D** B-cell responses were measured for young, low dose and old, high dose vaccination groups by measuring vaccine-specific IgGs in antibody secreting cells (ASC) using a CTL ELISpot scanner. The mean data is shown as a bar graph with error bars to indicate the SEM. Mean comparisons were performed by a 2-tailed paired Student's *t* test using GraphPad Prism software, but were not significant. The F Binding and ELISPOT analysis was performed at an earlier timepoint after three young and six aged biologically independent animal data had been accumulated. **E** Serum from day 35 (prime-boost) was used to determine neutralizing antibody titers using a luminescence-based microneutralization assay. Dot plots of neutralizing antibody titers, expressed as the reciprocal of the IC50 serum dilution with geometric means graphed and indicated as bars, were used to compare young, low dose vaccination vs old, low dose and old, high dose vaccinations. Statistical analysis was performed using a two-sample *t*-test (*p* = 0.041). Neutralization assays were run on duplicate plates and averaged prior to IC50 calculation with 4-month-old (young) and 17-month-old (old) male and female Balb/c mice totaling 12 or 13 biologically independent animals per group. For (**A**) to (**E**), source data are provided as a Source Data file.

individuals are often referred to as immunosenescent in that the overall responses to immune challenge are blunted due to age related factors. Overcoming this immunosenescence is critical to protecting this high-risk group. Several strategies exist to overcome immunosenescence via the dose, regimen, and/or adjuvant administered. We hypothesized that the improved potency of our molecule on alum would overcome immunosenescence simply with a higher dose avoiding harsher adjuvant and/or dosing requirements. As observed in Fig. 6E, while an equivalent dose given to old mice did not overcome

immunosenescence, simply increasing the dose by 4.5X elicited equivalent neutralizing antibody responses in young and old mice. Similarly, the high dose group demonstrated increased overall F binding antibodies that did not differ statistically from the young animals (Fig. 6C), and similar B-cell ELISpot responses were also observed in the old high dose group (Fig. 6D). These data demonstrate that the improved potency of the DT-preF molecule enables surmounting of immunosenescence with an equivalent vaccination regimen, with simply a higher dose. These data suggest that DT-preF will

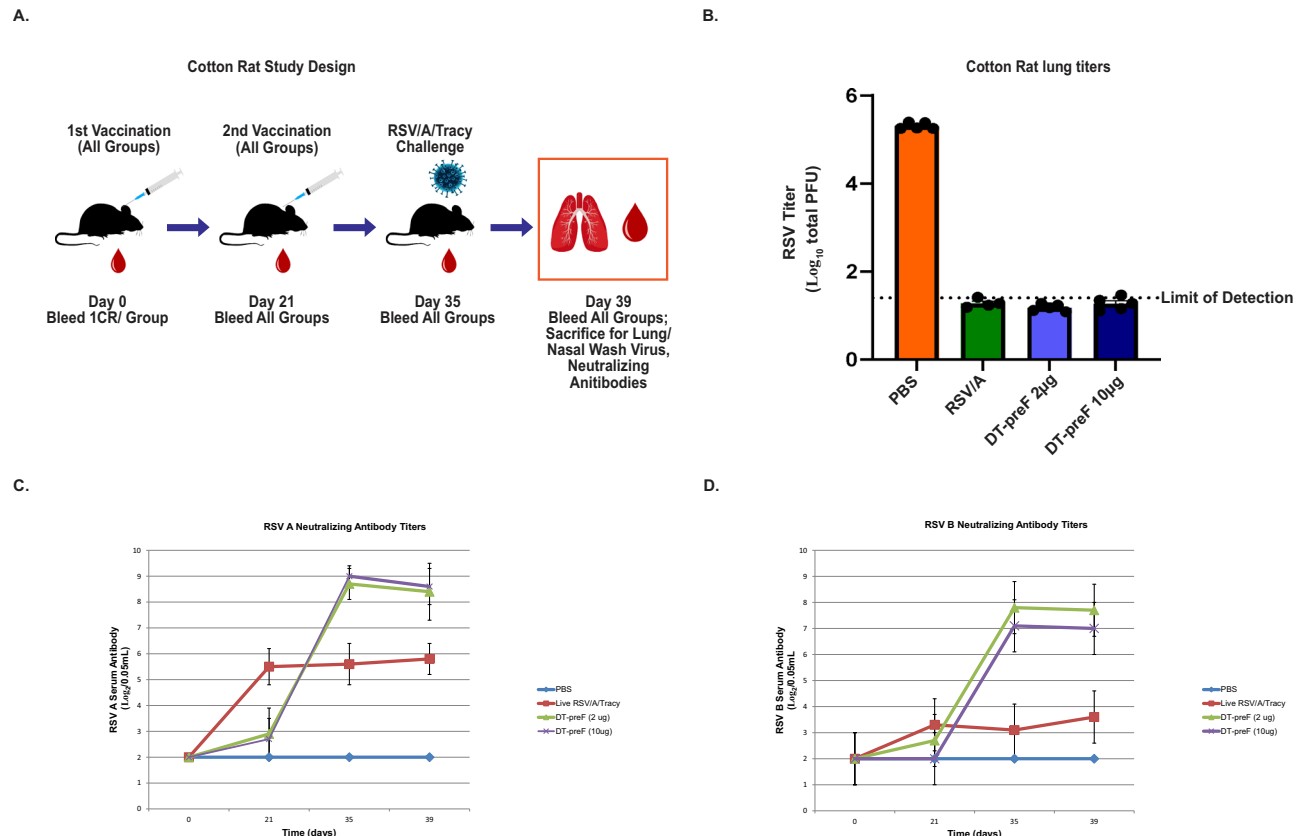

**Fig. 7 | DT-preF is a highly potent immunogen in cotton rats. A** Schematic describing the experimental design of the study. **B** Cotton rat lung titers were analyzed by plaque assay after vaccination with DT-preF (2 and 10 μg/animal) and the indicated controls at 4 days post challenge with $10^5$ pfu/animal RSV/A/Long. The limit of detection is indicated, individual animal values are plotted as dot plots, and the error bars represent the SEM. **C**, **D** RSV/A, and RSV/B cotton rat neutralizing antibody titers were measured in individual groups listed at the indicated time-points. Serum antibody levels were assessed up to 39 days, which includes day 4 post challenge. For all panels, $N = 5$ animals/group in a single cotton rat study. A two-tailed Student's t-test was used for statistical analysis. Minimal detection = 2.5. For (**B**–**D**), source data are provided as a Source Data file.

not require a less-tolerable adjuvant and/or repeat dosing to overcome immunosenescence.

## DT-preF is highly potent in cotton rats

We further tested the potency of our molecule in cotton rats to evaluate the responses, and the level of protection from challenge, that are elicited by low and high doses of DT-preF (2 μg and 10 μg doses per injection per animal) on alum. Controls included one group injected with saline (negative control) and one group was infected with live RSV/A/Tracy (positive control), as described in the materials and methods. Since animals were RSV naïve, a prime: boost regimen was used and animals were challenged on day 35 with $10^5$ pfu/animal of RSV/A/Tracy intranasally. The animals were sacrificed four days after challenge, and their lungs were collected for RSV titer analysis. The viral lung titers of both the 2 μg and 10 μg DT-preF groups of cotton rats were below the limit of detection (1.4 $\log_{10}$ pfu/g), whereas the saline control group had an average of $10^5$ pfu/g lung tissue (Fig. 7B) demonstrating that both doses of Alum-formulated DT-preF provided complete protection. Serum samples were collected on day(s) 0, 21, 35, and 39- to analyze NTs against RSV/A/Tracy and RSV/B/18537 subtypes; these results demonstrated that mean NTs of both DT-preF immunized groups exceeded those induced by live virus infection against both strains following the boost (Fig. 7C, D).

## Discussion

We have developed a technology to target dityrosine crosslinks into protein structures that can be applied broadly in protein engineering.

Herein, we describe its application to conformationally stabilize a viral fusion protein-based recombinant subunit vaccine, which in turn improves its potency, shelf-life, and cold-chain requirements. In addition to our stabilized RSV F protein, DT-preF, we are currently applying the technology to design and engineer a universal influenza vaccine immunogen.

Given the large market and medical need for an RSV vaccine, several groups are pursuing strategies encompassing multiple vaccine modalities (vector, mRNA, live-attenuated, nanoparticle, and their use in combination)[53–57]. The 2023 licensure of GSK and Pfizer's first generation subunit vaccines makes this a very exciting time in RSV vaccinology as first- in-class products will significantly improve the RSV public health crisis[32,58]. In RSV, subunit vaccines have demonstrated superb safety and efficacy, but instability of RSV subunit vaccines remains a key challenge for vaccine potency and delivery. We demonstrate that dityrosine bonding not only improves the stability of target molecules, but also significantly improves their potency.

In RSV, our current immunogen design introduces two dityrosine bonds into the F protein. The engineered inter-protomeric crosslink has been identified by LC-MS/MS(Fig. 3); no other crosslinks could be detected. But amino acid analysis confirms that more than one crosslink forms, and analytical SEC under denaturing conditions with in-line fluorimetry demonstrates the presence of an intramolecular bond (Fig. 4). The absence of nonspecific crosslinks detected by LC-MS/MS provides evidence for the specificity and selectivity of this targeted crosslinking technology. As demonstrated in Fig. 2C and Supplementary Fig. 2C, under typical reaction conditions, between 65–75% of the

protein contains all 3 inter-protomeric bonds and completely shifts to the trimeric species. However, the intramolecular bond forms to a lesser extent and reaches a total average of between 4 and 5 bonds per trimer supported by amino acid analysis. This suggests that some of the monomeric species contain the intramolecular bond. This is supported further by the HPLC analysis run under denaturing conditions.

We demonstrated that our crosslinked RSV preF immunogen has substantially better conformational stability than our comparator, DS-Cav1. Whereas after a 4-week period of storage at 4 °C DS-Cav1 elicited a roughly 60% drop in neutralizing Ab titers, neutralizing Ab titers elicited by DT-preF were not statistically reduced, demonstrating dramatically improved stability by targeted DT crosslinking. This scenario is particularly applicable to the distribution of a vaccine as it is expected to permit longer term storage of a multi-use vaccine vial, thus facilitating vaccine distribution. Additionally, dityrosine stabilization achieves 11x higher neutralizing antibody titers, through the elicitation of only 1.7x greater overall prefusion F binding titers. These data demonstrate that DT-preF elicits a higher percentage of neutralizing antibodies and thus a better-quality antibody response. The quality of the antibody response is particularly important when addressing safety of the immunogen since a high level of non-neutralizing antibodies were associated with VED in the initial FI-RSV vaccination trials[24,25].

A trend can be observed in the literature which suggests that engineering additional crosslinks to a prefusion pneumo and/or paramyxovirus subunit vaccine immunogen results in the elicitation of higher neutralizing antibody titers upon vaccination. This was shown in the case of the DS-2 molecule described by ref. 42, where the successful incorporation of an additional disulfide bond to DS-Cav1 achieved 4X greater neutralizing antibody titers. Similarly, Ou et al. reported a triple disulfide-bonded metapneumovirus F protein that achieved 10-fold higher titers than the single disulfide-bonded version of the protein[59]. There may, however, be a virus or protein-specific upper limit to potency improvements as incorporation of an additional disulfide bond in an otherwise single-disulfide bond stabilized parainfluenza virus type 3 fusion protein did not further improve neutralization titers[60,61]. Potency improvements are therefore more likely dependent on the ability of the additional crosslink to maintain the most authentic immunological shape rather than the numerical addition of crosslinks per se.

Overcoming immunosenescence in the 80+ and frail represents a critical unmet need in the current RSV first generation repertoire. While several levers can be used to surmount the immunosenescence challenge, starting with an immunogen with the highest potency provides an additional lever unavailable to the other immunogens. DT-preF's high potency overcomes immunosenescence in a mouse model solely with an increased dose on alum, an adjuvant known to be well tolerated. This demonstrates that DT-preF has an excellent chance of achieving improved protection in these most critical patient populations.

## Methods

### Cells, plasmids, antibodies, viruses
293 T cells were from an in-house stock originally from the Mount Sinai School of Medicine. 293 Freestyle cells were purchased from Invitrogen (ThermoFisher Cat#R79007) and cultured in 293 Freestyle medium (Gibco®). Hep-2 cells were purchased directly from ATCC (with a HeLa contamination disclaimer) (ATCC, Cat#CCL-23). Composite CHO cells expressing preF$^C$ were generated and cloned under contract with Abzena Ltd. preF$^C$ was cloned into the pcDNA 3.1 Zeo plasmid and used for transient transfections. Motavizumab, AM14, 5C4 anti-F primary antibodies were gifts from Jason McClellan (UT Austin) and Barney Graham, or purchased from Creative Biolabs (Shirley, NY). D25 and MPE8 were provided by Jason McClellan and Barney Graham. The anti-mouse and human secondary antibodies were purchased from GE Healthcare Life Sciences and Jackson ImmunoResearch Laboratories,

Inc. respectively. HRP Rat anti-mouse IgG2a (BD Pharmingen) was used in the conformational sandwich ELISA. Biotin-SP (long spacer) Affinipure goat anti-mouse IgG, Fc gamma fragment specific (Jackson Immunochemicals) was used in the ELISPOT assays. The RSV-luciferase virus was obtained from Dr. Martin Moore (Emory University).

### Protein expression and purification
Proteins were expressed by transient transfection of adherent HEK 293 T (screening), suspension HEK293F cells (production), or from a recombinant stable CHO cell line produced by contract from Abzena Ltd. (San Diego) expressing a tag-free version of preF$^C$(production). 293F Transfection was done at a cell density of $1 \times 10^6$ cells/mL in 1 L vented Erlenmeyer flasks with 1:3 DNA/PEI complex while 293 T cells for secondary (fluorescence) screening were transfected in a 96-well format with 250 ng of DNA, 1.5 μL PEI, and $3.8 \times 10^5$ cells/well. 293 cell supernatants were harvested after 4–5 days of expression and CHO cell culture supernatants were harvested from shake flasks over 14 days of expression. 293 F and CHO supernatants were centrifuged $2 \times$ (500 $g$ and 3100 $g$), passed over a cell strainer to remove any large debris, and purified by affinity chromatography over Strep-Tactin® Superflow® resin (IBA), or Capto-Adhere resin (Cytiva), respectively. 293F-expressed protein was eluted in a modified Strep-Tactin elution buffer (2.5 mM of d-Desthiobiotin (Sigma-Aldrich), 1 mM EDTA (Sigma-Aldrich), 1:20 of 100 mM Na Phosphate pH 8.4, in 1× PBS (LifeTech)) while CHO-expressed protein was eluted in a Citrate and L-arginine containing sodium phosphate buffer (pH 7.5). Eluted proteins were concentrated using Tangential Flow Filtration (TFF) (Minimate TFF Capsule with 50 kDa Omega Membrane, Pall Life Sciences) followed by Biorad, or total protein ELISA protein assays. Post DT crosslinking, in the case of 293-expressed DT-preF as well as Strep-Tactin purified DS-Cav1, further purification was done as follows: Imidazole at a final concentration of 20 mM was added to the reaction. To purify DT-preF from the crosslinking enzyme, ARP, the protein was purified over 5 mL HisTrap column (GE Life Sciences), using wash and elution steps composed of crosslinking buffer containing 150 mM salt plus 50 mM and 300 mM imidazole respectively. The proteins were further isolated using SEC. HiLoad Superdex 200 pg columns (GE Life Sciences) with PBS or 10%Sucrose/NTE as the elution buffers as indicated in the results section. DT-preF purified in 1× PBS was concentrated using Amicon Ultra-15 centrifugal filter units with an Ultracel-50 membrane (Sigma). CHO-expressed protein was purified after crosslinking by hydrophobic interaction chromatography using Butyl FF Sepharose resin (Cytiva) and used directly in the elution buffer for immunogenicity studies (10 mM Na-phosphate, 75 mM NaCl, pH7.0) or formulated as described in the stability study section.

When needed, in order to remove the affinity tags, 1 unit of biotinylated thrombin (Millipore/Sigma) was added to each 1 mL of concentrated SEC eluate (0.5–1 mg/mL) and incubated for 20 h at 4 °C. After the cleavage reaction, thrombin was removed according to the manufacturer's instructions followed by addition of 0.5 μL Strep-Tactin® resin per μg of protein cleaved, to bind/sequester the cleaved tags. Finally, the sample was spun in a tabletop centrifuge over a Pierce spin column (LifeTech) at 500 $g$ for 5 min to remove the beads and the protein concentration was measured by Ninhydrin assay[62]. The proteins were flash frozen in liquid $N_2$ and stored at −80 °C until further experimentation.

### ELISA-analysis
The initial expression-based screening of variants was done in a 96-well format ELISA, using supernatants from transiently transfected HEK293T cells. The cells were seeded in a 96-well tissue culture plate (VWR) at a density of $2.5 \times 10^4$ cells/well- on day 0 followed by overnight incubation. The plasmid DNA for the expression of designed variants was transfected 1:3 with Truefect Max (United Biosystem).

Supernatants were collected on day 5 and coated on Ni-NTA HisSorb plates (Qiagen). Motavizumab, AM14, MPE8, and D25 were used as the primary antibodies, at concentrations of 0.5 µg/mL, 1 µg/mL, 0.5 µg/mL, and 0.5 µg/mL, respectively in PBS with 20 mM Imidazole. Post-incubation and extensive washing, the plates were incubated with anti-human secondary antibody (Peroxidase AffiniPure F(ab')$_2$ Fragment Goat Anti-Human IgG, Jackson Immunochemicals, 1:2500) in PBS with 20 mM Imidazole. The plates were finally washed again, developed with 1-step TMB ELISA substrate solution (Thermo) and read in a NOVOStar plate reader.

Antibody responses in serum were quantified by standard ELISA methods, with DS-Cav1 immobilized on the plate as the capture antigen. HRP-labeled anti-mouse IgG (GE Healthcare) served as the secondary antibody and One-step TMB (Invitrogen) was the developing reagent. Binding titers are shown as the reciprocal of the serum dilution resulting in an OD$_{450}$ value of 0.75 to lie within the linear range of all serum binding curves.

Antibody responses in aged animals were conducted by standard ELISA methods with DT-preF immobilized on the plate as the capture antigen. Sera from different timepoints were tested (one day before vaccination for pre-vaccination values (t0), boosted 21 days (t21) and 35 days (t35) after prime/boost). 96-well plates were coated with 2 µg/ml of DT-preF for 1 hr, at RT, washed and blocked with 1%BSA for 30 min, at 37 C, after washing, sera was diluted at 1:20000 and added for 2 h at RT, wells were washed and anti-mouse IgG-HRP-conjugated (Jackson 115-036-062) was added for 1 h at RT, after washing, TMB substrate reagent (BD 555214) was added to all wells. Plates were read for absorbance at 450 nm (BIO-RAD Benchmark Plus microplate spectrophotometer).

### SDS-PAGE analysis

During preliminary screening, inter-protomer crosslinks in Streptactin-purified variants were assessed using SDS-PAGE gel shift. Uncrosslinked and crosslinked protein samples were mixed 1:4 with NuPAGE 4X LDS sample buffer (LifeTech). TCEP (Sigma-Aldrich) was used 1:10 as the reducing agent. Samples were then cooked at 95 °C for 5 m. After cooling, samples were loaded and run in concentration-compatible volumes on a 4–12% Bolt mini gel (1 mm × 12-wells; LifeTech), followed by staining with Quick Coomassie Stain (Anatrace).

### DT-crosslinking reaction/fluorescence analysis

For screening, 293 T transfected cell supernatants were clarified by centrifugation, diluted 1:1 with PBS, normalized for total protein by Motavizumab ELISA, and crosslinked through the addition of Arthromyces ramosus peroxidase (ARP) at 11 µg/mL. The reaction was initiated with the addition of 0.00036% hydrogen peroxide (H$_2$O$_2$; Sigma-Aldrich) at room temperature. Fluorescence intensity measurements were taken prior to the initiation of the reaction and post 15 min of crosslinking in a Novostar plate reader with an excitation wavelength of 320 nm and an emission wavelength of 405 nm. For large scale protein preparation, the reaction was conducted similarly in 30 mL volumes in Falcon tubes or using a DASGIP parallel bioreactor system in a 1 L BioBlu 1c single-use vessel at 39–43.0 °C. Crosslinking was conducted to specification using either ARP enzyme or a Nickel-peptide complex as the catalyst and hydrogen peroxide or magnesium monoperoxyphthalate as the oxidant. 150 mM NaCl was added to the crosslinking buffer and the reactions proceeded for 5–50 min.

### Mass spectrometry analysis

Endopeptidase trypsin (modified, sequencing grade) was purchased from Promega (Madison, WI). Chymotrypsin was obtained from Roche (Indianapolis, IN). The Quadrupole ion trap mass spectrometer (LTQ XL) used in the proteomic analysis was manufactured by Thermo (Palo Alto, CA). After sample denaturation, reduction, and alkylation, samples were solution digested for LC/MS/MS sequencing with the following digestive enzymes: trypsin, chymotrypsin, and pepsin. The digested peptide mixture was analyzed by an LC-ESI-MS/MS system, in which an Agilent 1100 Binary pump high-performance liquid chromatography (HPLC) system was used to run a 75 micrometer inner diameter reversed phase C18 column which was on-line coupled with an ion trap mass spectrometer. The solvents used for HPLC were solvent A 98% H2O, 2% acetonitrile and solvent B 10% H2O, 90% acetonitrile, both contain 0.025% TFA. A combination linear/step gradient elution was performed between 2 and 90% solvent B over 180 min as outlined in the Supplementary materials and the analysis time was 200 min. Peaks were manually assigned using Prottech's in-house software (proprietary), while data for presentation was prepared with Thermo XCalibur Qual Browser 3.1 Release. Mass spectrometry data for the crosslinked and uncrosslinked proteins have been deposited to the ProteomeXchange Consortium via the PRIDE partner repository with the dataset identifier PXD048897.

### Amino acid analysis

Crosslinked protein, a chemically synthesized dityrosine standard (Chemshuttle), and commercially purchased tyrosine standard (Sigma) were shipped frozen to the University of California, Davis molecular structure facility. A precise volume/mass of protein was transferred, dried, and hydrolyzed in 6N HCL, 1% Phenol, 110 °C, 24 h, in vacuo. Samples were then dissolved in a precise volume of sodium diluent (Pickering Labs) containing 40 nmol/mL NorLeucine. 50 µl was injected on a Concise (AminoSep Beckman Style Na+ column, part #AAA-99-6312) strong cation exchange column using a Na-based Hitachi 8800. An amino acid standards solution for protein hydrolysate (Sigma, A-9906) was used to determine response factors, and thus calibrate quantities for all amino acids. Each injection also contains norleucine as an internal standard to allow correction of the results for variations in sample volume and chromatography variables. Data was reviewed by two staff members for accuracy.

### Analytical HPLC-size exclusion chromatography

Crosslinked and uncrosslinked protein samples were treated with Remove-iT PNGase F (New England Biolabs) according to the manufacturer's instructions, followed by denaturation with SDS. The denatured protein was reduced and alkylated with the G-biosciences FOCUS™ protein reduction-alkylation kit according to the manufacturer's instructions (St. Louis, MO). Samples were run on an Agilent 1260 Infinity II HPLC system using two LW-403 4D (Shodex) analytical size exclusion columns in tandem at 45 °C with a mobile phase of 0.2% SDS, 50 mM NaH$_2$PO$_4$, 200 mM NaCl (pH 5.5).

### Stability assay/ELISA

DT-preF and DS-Cav1 purified proteins were thawed on ice followed by dilution to 100 µg/mL (Ni-NTA) or 41.5 µg/mL (sandwich/polysorp) with a final 5% sucrose buffer concentration. PMSF at a final 1 mM concentration was added and the proteins were stored at 4 °C until the stability ELISA timepoints indicated. To confirm equal protein in all samples at the onset of the study, Motavizumab ELISA was performed. Proteins from the same stocks were freshly thawed on ice at each timepoint followed by equivalent dilution and PMSF treatment to serve as "Fresh" control samples. The freshly thawed and 4 °C stored samples were compared for antigenicity on microtiter 96-well Ni-NTA HisSorb plates (Qiagen) or on 96-well Immuno Polysorp plates (Thermo Fisher Scientific) for total protein ELISA and microtiter 96-well Ni-NTA HisSorb plates (Qiagen) for AM14 conformational ELISA or 96-well high binding EIA/ RIA plates for 5C4/ AM14 conformational sandwich ELISA. The antigen was coated on plates for 2 h at RT or at 37 °C for 1 h 40 m for total protein and AM14 ELISA and overnight at 4 °C on AM14 coated plates for the conformational sandwich ELISA.

The plates were then probed with primary antibodies Motavizumab (at 0.5 μg/mL), AM14 (1 μg/mL, Ni-NTA ELISA) or 5C4 (2.5 μg/mL, 5C4/AM14 sandwich ELISA) or D25 10 μg/mL by incubation for 1 h at RT or 37 °C 1 h, respectively. HRP-conjugated secondary antibodies (Amersham ECL HRP Conjugated Antibodies 1 ml sheep anti Mouse, Peroxidase AffiniPure F(ab')₂ Fragment Goat Anti-Human IgG, Fcγ fragment specific and HRP Rat Anti-Mouse IgG2a) were then added for a 1 h RT or 37 °C incubation, respectively. The assay was developed using HRP colorimetric substrate 3,3′, 5,5′-tetramethylbenzidine (TMB) substrate. Plates were read for absorbance at 450 nm after sulfuric acid addition.

## Murine vaccinations

Animals were handled according to the IACUC protocols at the University of Miami (Protocol #19-013-LF) and SUNY Downstate (Protocol #16-105-12), respectively. For the stability study in Fig. 5B, 6–8 week-old, female CB6F1/J mice and the adult vaccination study in Fig. 6A, B, 15-week-old, female CB6F1/J mice were vaccinated by IM injection into the hindlimb (50 μl) with either 10 μg (stability) or 2.3 μg (potency) of DT-preF or DS-Cav1 formulated as indicated (Advax^SM 1 mg/mouse (Vaxine pty), 2% Alhydrogel (Brenntag, 100 μg/mouse)). Animals were primed on day 0, boosted on day 21, and serum was collected on day 35 for determination of neutralization titers/prefusion F binding titers. For the aged animal vaccinations, 17 months old male and female BALB/cJ mice were primed (day 0) and boosted on day 21 via IM injection (50 μl) into their right hindlimb with vehicle (PBS), or DT-preF (10 μg, low dose, or 45 ug, high dose) formulated with Alhydrogel as described above. Mice were bled one day before vaccination for pre-vaccination values (t0), at 21 days (t21) prime only. All mice were sacrificed at 35 days (t35) after prime/boost. There were a total of 12 or 13 animals in each group.

## Neutralization assay

Heat-inactivated serum is diluted 1:50–1:800 (depending on the expected potency of the serum) in phenol-free MEM supplemented with glutamax, 5% FBS, and Penicillin/Streptomycin and serially diluted in a 3-fold series (50 μL final volume). RSV-Renilla Luciferase virus was then diluted in the same medium as above to $2 \times 10^4$ pfu/mL and 50 μL is added to each well (10,000 pfu/well) with virus only controls. Virus and serum are incubated at 37 °C, 5% $CO_2$, for 2 h. Hep-2 cells (ATCC) are then trypsinized, counted, and diluted to $1 \times 10^6$ cells/mL in the same medium and 25 μL is added to each well ($2.5 \times 10^4$ cells/well). The cells are then incubated at 37 °C, 5% $CO_2$ for 60–72 h and luciferase activity is quantified using the Renilla-glo luciferase assay system (Promega) according to the manufacturer's instructions and read in a Novostar plate reader. Data is analyzed using nonlinear regression to calculate $IC_{50}$ concentrations of each serum dilution curve (GraphPad Prism).

## B-cell ELISPOT analysis

96-well plates were coated with DT-preF (2 μg/ml), o/n 4 °C, washed, blocked with 1% BSA and then incubated at 37 °C for 12–18 h with culture-generated memory B cells from Day 35 spleens, with a 2- fold serial dilution starting at $2-4 \times 10^6$. Plates were then washed and incubated with Biotin-goat anti-Mouse IgG Fc specific (Jackson 115-065-071), 1 h RT. Plates were then washed, streptavidin-HRP (Jackson 016-030-084) was added, and the reaction was developed with AEC substrate (BD 551951). Plates were scanned and analyzed with a CTL ELISpot Scanner.

## Cotton rat studies

To determine neutralization titers elicited by injection of DT-preF vs. a live virus infection control in the cotton rat model of RSV challenge, a prime-boost immunization scheme was utilized. Experiments were performed utilizing NIH and United States Department of Agriculture guidelines, The Public Health Service Policy on Humane Care and Use of Laboratory Animals, and experimental protocols approved by the Baylor College of Medicine's Investigational Animal Care and Use Committee (IACUC; Protocol # AN-2307). $10^5$ PFU/rat of virus (RSV/A/Tracy for Fig. 7) was intranasally administered to lightly anesthetized animals in the live RSV groups only on day0, and for all animals on the day of challenge (day 35 or 49).

5 cotton rats/group (roughly equivalent male:female ratio) were injected intramuscularly with DT-preF at 2 μg or 10 μg plus Alhydrogel 2% (ALH; Brenntag). The second vaccination (boost) was given on day 21 followed by intranasal challenge on day 35. On day 39 (4 days post challenge), animals were euthanized, and the lungs were harvested for determination of viral lung titers and histopathology according to established protocols. On all indicated days (day 0, day 21, day 35, and day 39) blood was obtained for serum analysis including microneutralization (Nt) assays for serum neutralizing antibodies to RSV/A/Tracy and RSV/B/18537 performed with HEp-2 cells following heat-inactivation at 56 °C for 30 min. Serial two-fold dilutions in duplicates starting at 3-log₂ were performed to determine the neutralizing antibody (Ab) titer for each sample. The neutralizing antibody titer is defined as the serum dilution at which ≥50% reduction in viral cytopathic effect (CPE) is observed. As an internal standard, Palivizumab was included at 40 μg/mL.

## Statistical analyses

For neutralization titer analysis, groups of animals were compared on the distribution of the neutralization titer outcome using the Mann-Whitney test (Fig. 6A) or two-sample $t$-test (Fig. 6E). Groups consisted of ten animals except one animal that was excluded from the DT-preF group in the potency study since it was found dead before the onset of vaccinations. GraphPad Prism and Excel were used to conduct statistical analysis. $P$-values are two-sided and are considered significant at a one-sided 0.05 level of significance. For the F binding and B-cell ELISPOT analyses, mean comparisons were performed by a 2-tailed paired Student's $t$ test using GraphPad Prism software. The F Binding and ELISPOT analysis was performed at an earlier timepoint after three control and six aged animal data had been accumulated. Full neutralization titers were performed with male and female animals totaling 12 or 13 per group. Cotton Rat analysis was performed with male and female animals (5 per group). Data was analyzed using a two-sided student's $t$-test with a $p \le 0.05$ level of statistical significance.

## Reporting summary

Further information on research design is available in the Nature Portfolio Reporting Summary linked to this article.

## Data availability

All data that support the findings of this study are available in the source data file. The constructs designed in this study are based on the GenBank sequence: 5W23_A and the crystal structures used for construct design are fully described in the cited references. Mass spectrometry data for the crosslinked and uncrosslinked proteins have been deposited to the ProteomeXchange Consortium via the PRIDE partner repository with the dataset identifier PXD048897. Source data are provided with this paper.

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

## Acknowledgements

We would like to thank the lab members at the IAVI DDL for sharing numerous reagents and equipment. We would like to thank the members of Prottech Inc for performing the mass spectrometry analysis, the HAN Biocentre (Nijmegan, NL) for providing the ARP, and Creative Biolabs for performing DSF on the crosslinked molecule. We would like to thank Nikolai Petrovsky for providing the Advax<sup>SM</sup> adjuvant. Statistical analysis consultation was kindly provided by the late Joe Massaro, PhD (Boston University) through Dyad Systems consulting. We would also like to thank our collaborator Barney Graham, M.D. Ph.D. (NIH-VRC), for providing critical antibodies and helpful discussions, the members of Chemitope glycopeptide for helpful discussions, and the team at Baylor School of Medicine who performed the cotton rat study. We thank the members of our Scientific Advisory Board members Florian Schodel, M.D. and Barry Buckland, Ph.D., for their support and input and our manufacturing consultant, Beth Junker, Ph.D. for technical expertize and advice. We also thank Claudio Bertuccioli, Ph.D., the late Neil Kreiger Ph.D., and Maria Paikos-Hantzis B.S., for assistance and helpful discussions. We also thank Evgeny Vulfson for early work with the dityrosine crosslinking reaction. This work was funded by NIH grants: 1R43AI112124 (RM/JM/MY), 1R44AI112124 (MY), R44AG064107 (MY) and we would like to thank our program officers Sonnie Kim, (AI) and Rebecca Fuldner, Ph.D. (AG). This work was also funded by the NY state Biodefense fund awards over 2 years and would like to thank the administrators of this program. We also thank the IndieBio NY program and Stephen Chambers for helpful deliberations.

## Author contributions

The project was conceptualized by C.M. and R.M., and constructs were designed by C.M. and J.S.M. R.M. initially, followed by M.A.Y. designed and directed the project. S.G. and D.B., served as principal scientists on the project and were involved in planning and conducting the majority of the experiments resulting in the generation, analysis, and testing of the immunogen. A.Z., M.B., J.Z., R.M., P.A., B.B., D.F, J.M. and M.A.Y., also planned and conducted experiments resulting in the generation, analysis, and testing of the immunogen. J.S. performed the amino acid analysis. The paper was written by M.A.Y., S.G., M.B. and C.M. with editing comments provided by most authors.

## Competing interests

J.S.M., R.M., P.A., C.P.M. A.Z, D.B., S.G., and M.A.Y. are named as inventors on patents and patent applications related to the data presented in this work. S.G., D.B., A.Z., J.S.M., R.M., P.A., C.P.M. and M.A.Y. are shareholders of Calder Biosciences Inc. The authors declare no other competing interests.
