## [Peer Review File · Nature Communications]

Engineered dityrosine-bonding of the RSV prefusion F protein imparts stability and potency advantagesReviewers' Comments:

Reviewer #1:

Remarks to the Author:

Engineered dityrosine-bonding of the RSV prefusion F protein imparts stability, safety, and potency advantages.

Gidwani et al.

The main goal of the study is to produce a stabilized RSV F protein in the pre-fusion conformation that can increase the shelf time, while remaining as an active, high quality prefusion form of RSV F antigen, for vaccine production. The group has been precursors of using tyrosine crosslinking to generate dityrosine bonds and stabilize the protein. The study shows first the screening of mutations in order to pinpoint the best antigen (without modifying its antigenic activity compared to the precursor DS-Cav1), characterize the protein generated, and test the protein in conditions of lengthy storage at 4oC. Finally, immunogenicity studies are performed in mice and in cotton rats to show its immunogenicity advantages.

Overall, the manuscript lacks experiments and data key to generate confidence that the antigen produced (DT-PreF) is definitively superior in all aspects indicated, to DS-Cav1. The data of stability is weak and incomplete, the safety data is not shown, and the potency data presented is seen as only modest and with gaps that preclude clear conclusions of why this is better than DS-Cav1.

The protein needs more convincing characterization. In fact, in the X-axis for the graphical data in figure 1, it is evidenced that there are other prototype mutations that give equal or better binding activity to important antibodies (D24 and AM14) than the one chosen. However, no explanation of why these constructs weren't further analyzed.

Another weak point is the absence of binding data and comparison with DS-CAV1 of other important conformation-recognizing antibodies as they are D25 (site θ , left outside in figure 2), as well as sites I and III (also precluded in the post-fusion F form).

In figure 3A, DS-Cav1 should be included as control for the analysis, with and without cross-linking.

The data on stability is not compelling. Figure 4A should also assay for D25 binding. Figure 4B, in addition to the percentage potency loss, the supporting neutralization data should be included, with serum samples before and after booting, and indicating the number of animals used for each antigen. Statistical analysis should be provided.

Figure 5A, the comparison immunogenicity in the BALB/c model, is not particularly robust, and lack of statistical analysis, as well. Figure 5B and Figure 6 do not show comparison with DS-CAV1, required for superiority analysis.

Although the manuscript's title mentioned safety, no data is shown.

Reviewer #2:

Remarks to the Author:

Gidwani et al. explore the ability of targeted dityrosine crosslinks to stabilize the RSV F glycoprotein in its prefusion conformation. The authors start with RSV F A2 strain as well as with the two Cav1 mutations that enhanced the prefusion stability in the DS-Cav1 variant of RSV F. They design several dozen dityrosine variants capable of forming intra or interprotomer dityrosine linkages, assessing these for recognition by the motavizumab antibody as well as the prefusion specific antibodies D25 and AM14. The top 16 were expressed, purified, and subjected to crosslinking, with one variant, called

DT-preF, with two engineered dityrosine bonds, V185Y pairing with N428Y, and K226Y pairing with the endogenous tyrosine at Y198, appearing especially stabilized.

The formation of dityrosine crosslinks in DT-preF was partially confirmed by mass spectrometry identification of one of the crosslinks, with amino acid analysis indicating the presence of a third dityrosine crosslink in each monomer, and the crosslinked protein assessed for stability at 4 C and found to be substantially more stable after 5 weeks, with immunogenicity assessment after 4 weeks of incubation showing substantial reduction in elicited neutralization titers for DS-Cav1 but not DT-preF.

DT-preF was tested versus DS-Cav1 in naïve mice, which at 1 ug/ml injection in Alum, show a 9-fold increase in relative immunogenicity. The ability of DT-preF to elicit titers in aged mice was also assessed, with increased dosing enabling similar elicited titers in aged and young mice. DT-preF was also tested in cotton rats and found to be highly potent at reducing lung titers in RSV challenged animals. Overall, the authors use dityrosine crosslinking to stabilize RSV F in its prefusion conformation and show the resultant immunogen (DT-preF) to have improved stability and immunogenicity.

It's exciting to have a new type of structure-based stabilization – enzymatically induced dityrosine crosslinks – to fix the conformation of an important vaccine antigen, and the data do suggest DT-preF is more stable and elicits improved immune responses versus DS-Cav1 in naive cotton rats and mice, the latter assessed in both standard and aged mice.

That said, several aspects of the manuscript, including referencing, described data, and order of experiments, need to be improved.

The referencing is a mess. For example, it's helpful for the authors to alert readers to the multiple prefusion stabilized RSV Fs immunogens that are in phase 3 development, such as from GSK, Pfizer, J&J and Moderna, and we are told about references 38 and 39 that (lines 83-84) that "Current RSV prefusion stabilized first generation vaccines have built upon the success of DS-Cav1 and have cleared phase 3 clinical development with licensure expected in the coming months (ref. 38,39)." However, reference 39 refers to an influenza paper (Sahni et al. Sustained Within-season Vaccine Effectiveness Against Influenza associated Hospitalization in Children: Evidence From the New Vaccine Surveillance Network, 2015-2016 Through 2019-2020. Clin Infect Dis 76, e1031-e1039, 597 (2023).) And while Ref. 38 (Papi et al.) is reasonable, the authors should cite other Phase 3 results such as from J&J (e.g. Kampmann et al. NEJM 2023), along with references to Pfizer's and Moderna's efforts. In terms of other prefusion-stabilized RSV F, "Several 2nd-generation RSV subunit vaccines are being developed that further improve upon DS-Cav1's stability (ref. 32-37).", the authors should cite Joyce et al., Iterative structure-based improvement of a fusion-glycoprotein vaccine against RSV. Nat Struct Mol Biol. 23, 811-820 (2016). This reference describes variants of RSV F that have substantially improved stability and 4-5-fold improved immunogenicity versus DS-Cav1 and are thus likely the closest 2nd-generation version of RSV F to DT-preF. In addition to checking citations, the authors should discuss how their DT-preF compares to these 2nd generation variants.

Critical data is not provided. Since DT-preF is the "new" immunogen that is the focus of the paper, it's critical to provide data on its development and to characterize the two dityrosine crosslinks. In terms of development, it's nice to see that antigenic data on V185Y pairing with N428Y in Fig. 1B, but nothing is shown for this variant in terms of DT fluorescence (Fig. 1C). Also, the rationale for combination is unclear; many combinations are possible, yet no mention is made of how the specific pair that makes up DT-preF was chosen. In terms of characterization, two aspects are critical, whether the crosslinks actually form and their frequency. For formation, Fig. 1C shows the fluorescence from the purported crosslink between K226Y pairing with the endogenous tyrosine at Y198 to be one of the lower reported values, less than half that for some of the other crosslinks. For frequency, we can see substantial monomers and dimers on the SDS-PAGE of DT-preF (Fig. 2C). It appears from this analysis that only about half the intramolecular crosslinks form (it would be good, if the authors were to scan

the gel and to report the quantification of monomer, dimer, and trimer components, and along with the frequency of intramolecular crosslink). We are told, moreover, that for the critical experiment confirming the formation of a crosslink between K226Y pairing with the endogenous tyrosine at Y198: "Amino acid analysis confirmed dityrosine's presence in the sample and between 3 and 4 dityrosine bonds were present on average per molecule (data not shown)." Please show this data. Also, explain how this data is consistent with the SBS-PAGE analysis indicating that a substantial fraction of intermolecular crosslinks do not form. Overall, the authors need to show data that prompted their selection of the crosslinks in DT-preF and to define (or at least estimate with appropriate uncertainty) the frequency of their formation.

Order of experiments should be improved. Figure 4 is out of place. While I like that the authors test "shelf-life" by assessing immunogenicity from 4 C stored immunogens, I found this description prior to the characterization of relative immunogenicity in Fig. 5 to be suboptimal. Rather, it would be better to place the combined immunogenicity-stability analysis after the first description of immunogenicity. Also, it's strange to report only the ratio of the elicited neutralizing titers, not the actual neutralizing titers themselves, please report actual neutralization titers for DT-preF and DS-Cav1 shown in Fig. 4 (for both fresh and after 4 weeks at 4 C).

Other aspects of the paper that should be improved:

1. The RSV F is a glycoprotein.
2. The authors utilize Cav1 mutations to help stabilize. I assume this is because the Cav1 mutations provide some prefusion stability, which is need during purification, prior to enzymatic dityrosine crosslinking. The authors should discuss whether experiments would have been successful without Cav1 and what would have happened if they introduced the dityrosine crosslinks on DS-Cav1.
3. In Fig. 1B and 1C, the order of the variants on the X-axis appears to be random, making it difficult to find data on specific variants tested. Please arrange in a clear order; I suggest numerical order by residue number.
4. In Fig. 1C, the authors should clarify protein quantification, and provide fluorescence as normalized by the amount of protein.
5. The authors should provide the fluorescence of DT-preF, as normalized by the amount of protein.
6. There appear to be some typos in variant names in Fig. 1 (e.g. L18Y8 in panel C).
7. In Fig. 5, please provide an actual number for geometric mean immunogenicity (with error) for panels A and B.
8. In the methods, it says 10 mice/group, but there are only 9 values shown for DT-preF in Fig. 5A/B.
9. Since this is one of the first papers on designed dityrosine crosslinks, it would be helpful for the authors to discuss what aspect of the design correlated best with formation of crosslinks.g
10. In terms of the title, the authors report stability and potency advantages, but the safety advantage should be clarified.

Reviewer #3:

Remarks to the Author:

In this work, the authors use protein engineering to stabilize a viral fusion protein in its pre-fusion

conformation, with the goal of developing an improved vaccine. The key outcome of this work seems to be improved storage stability of this modified protein complex. While I am not an expert on vaccine development, the authors do show results of a rat study in Figure 6, and these indicate efficacy. Results from mouse serum in Figure 5 further seem to support this, although the difference in neutralization titer between this new construct and the comparator DS-Cav1 apparently barely reaches significance ($0.01 < p < 0.05$). Still, the emphasis is on storage stability, so increased efficacy versus the comparator seems to be less critical.

As my expertise is primarily in mass spectrometry, I focused on this aspect during my reading. Overall, this work seems interesting, and the protein MS results in Figure 3 are reasonably convincing, although I would prefer to see more data, as well as more experimental detail.

*The description of the RP-HPLC separation should be expanded. Currently, the text starting on line 357 does not even specify the HPLC instrument used (it might be the same one as for the SEC experiment, but this is not stated). The LC gradient is also not described, beyond a statement that the total analysis time was 200 minutes. MS data analysis (in particular, which software was used) is also not described in any detail.

*The text states (line 143) that 'Unique peptides were identified in the crosslinked sample, and these were selected and sequenced by tandem MS.' Figure 3, however, only shows tandem MS results for a single peptide – was this the only one that was identified? How many other unique peptides were detected that could not be identified?

*Related to the above point, the mass errors in the spectrum shown in Figure 3A seem to be quite large – I did not check all of the m/z values, but found several that seem to be off by over 100 ppm, especially at higher masses. While it does seem likely that peptide X was assigned correctly based on the spacing of the fragment signals, there do not seem to be any fragments that specifically confirm the identity of the covalently bound peptide Y (other than a possible signal of the intact peptide Y at 395.3 m/z). I appreciate that this is a targeted experiment and therefore lower instrument performance characteristics can be tolerated than in most proteomics studies. Even so, this is a sizable dataset and I would still be worried about false-positive assignments, especially if a large mass error is tolerated and if this was the one peptide (out of an unknown number of unique ones) where a plausible interpretation was found. Do the authors have additional evidence for this peptide identification?

*Did the authors also process the peptide signals that were not unique to the crosslinked sample? If yes, what sequence coverage was obtained? I realize that it's potentially some extra work to revisit the dataset and investigate this, but it could provide insight into the overall data quality, including typical mass errors of the non-crosslinked fragments.

*It would be helpful to interested readers if the MS data were made available in a repository such as PRIDE, as recommended in the editorial policies (<https://www.nature.com/nature-portfolio/editorial-policies/reporting-standards>) and the identifier provided in the 'Data availability' statement.

*Minor point: in Figure 3A, it is unclear which signal the arrow for fragment Xb8 (which I would expect at 729.4 m/z) is pointing at.

We would like to thank all the reviewers for their thoughtful review of our manuscript. Their critiques have undoubtedly strengthened the manuscript significantly and we have paid careful attention to address the concerns raised. Included below is a summary of changes that we made to address the concerns. Please note that all line numbers correspond to the “track changes” version of the manuscript.

Reviewer 1:

Overall, the manuscript lacks experiments and data key to generate confidence that the antigen produced (DT-PreF) is definitively superior in all aspects indicated, to DS-Cav1. The data of stability is weak and incomplete, the safety data is not shown, and the potency data presented is seen as only modest and with gaps that preclude clear conclusions of why this is better than DS-Cav1.

We thank the reviewer for their careful review and concerns. We have generated a significant amount of data for the resubmission that we feel should assuage their concerns. Specifically, we have eliminated safety from the title and performed a new stability study which includes data out to 11 weeks of storage at 4°C and includes D25 analysis in addition to the highly prefusion specific AM14 and 5C4 antibodies used in a sandwich ELISA format. We have also added differential scanning fluorimetry analysis and demonstrate a 25°C shift in the melting temperature of the crosslinked molecule (Supplemental figure 3A). Additionally, we have obtained the highest reported potency in the field (11.1X the DS-Cav1 comparator, Figure 6A) with solely a prime:boost regimen and have obtained a p value of 0.01 to address the reviewer’s potency concerns. The highest potency previously reported (9X DS-Cav1, Marcandalli et al. Cell 2019) was achieved only through the use of a non-conventional, 3 vaccination regimen (prime:boost:boost).

The protein needs more convincing characterization. In fact, in the X-axis for the graphical data in figure 1, it is evidenced that there are other prototype mutations that give equal or better binding activity to important antibodies (D24 and AM14) than the one chosen. However, no explanation of why these constructs weren’t further analyzed.

We agree that our description of the selection criteria was confusing. Many factors are included in the lead molecule selection process and we agree that the explanation for our selection of DT-preF should be more descriptive. In this vein, we have incorporated additional text to address this shortcoming including descriptions of selection criteria in the body text on lines 101-105 and in the results section on lines 135-137, 149-161, 164-165, and 170-171 (all described in the track changes version).

Another weak point is the absence of binding data and comparison with DS-CAV1 of other important conformation-recognizing antibodies as they are D25 (site θ , left outside in figure 2), as well as sites I and III (also precluded in the post-fusion F form).

In order to address this concern, we have performed direct binding ELISAs using all of our available antibodies including D25, MPE8, and Motavizumab (Supplementary Figure 1A). The manuscript already heavily relied on AM14 and we have now incorporated both AM14 and 5C4 into a highly prefusion-specific ELISA format as demonstrated in our expanded stability figure (Supplementary Figure 3B). We have also substituted our screening ELISA Figure to incorporate additional antibodies as suggested (Figure 1B).

In figure 3A, DS-Cav1 should be included as control for the analysis, with and without cross-linking.

Cav1 represents the best control for our molecule since we built upon this molecule for our DT substitutions. We therefore focused most of our response on the Cav1 control. Nevertheless, to address this concern, we have exposed both the DS-Cav1 and Cav1 molecules to the crosslinking conditions and show no intermolecular bond formation as apparent by MW shift under denaturing conditions (Coomassie gel of reactions (Supplemental Figure 2A). We further isolated and purified the Cav1 molecule to provide additional support through Coomassie/Western blot analysis and HPLC run under denaturing conditions (Supplemental Figure 2b). Given these data and the costs involved with conducting the type of mass spectrometry analysis represented in this manuscript we did not pursue these controls further. We have added text describing these

results in the introduction and results sections (lines 101-105 and 195-207) to clarify the specificity of the reaction.

The data on stability is not compelling. Figure 4A should also assay for D25 binding. Figure 4B, in addition to the percentage potency loss, the supporting neutralization data should be included, with serum samples before and after boosting, and indicating the number of animals used for each antigen. Statistical analysis should be provided.

We have expanded our stability analysis to include the neutralization titers of each animal and statistical analysis in the new Figure 5 according to the reviewer's comments and have added a supplementary figure that addresses the reviewer's D25 concerns (Supplemental Figure 3B). Included now is a stability study which includes data out to 11 weeks of storage at 4°C and includes D25 analysis in addition to the highly prefusion specific AM14 and 5C4 antibodies used in a sandwich ELISA format. We have also added differential scanning fluorimetry analysis to this supplemental figure. We do not, however, collect serum prior to the boost since the animals are naïve and require a prime:boost regimen to show measurable responses in our hands.

Figure 5A, the comparison immunogenicity in the BALB/c model, is not particularly robust, and lack of statistical analysis, as well. Figure 5B and Figure 6 do not show comparison with DS-CAV1, required for superiority analysis.

Aged animal studies are particularly time consuming, laborious, and expensive. We designed and conducted these studies to optimize adjuvant selection/vaccination for our DT-preF molecule with these concerns in mind and therefore the control for our immunogen in these studies was young mice, injected with a low-dose of immunogen. Using this design, we were able to identify a vaccination formulation/regimen in aged mice that elicits equivalent titers to the young mice given a low dose of vaccine. This achievement stands alone and we do not compare, or claim superiority to, DS-Cav1 in this part of the study. We have added statistical analysis to the new Figure 6E as requested. Our cotton rat study was conducted under contract from the NIH and did not include a DS-Cav1 comparison group. Nevertheless, we do not claim superiority in this model in the manuscript and it still represents critical data needed for an IND submission since the control in this study was live virus infection.

Although the manuscript's title mentioned safety, no data is shown.

We have modified the title of the manuscript and eliminated all relevant instances of the word "safety" to address the reviewer's concerns.

Reviewer #2 (Remarks to the Author):

Gidwani et al. explore the ability of targeted dityrosine crosslinks to stabilize the RSV F glycoprotein in its prefusion conformation. The authors start with RSV F A2 strain as well as with the two Cav1 mutations that enhanced the prefusion stability in the DS-Cav1 variant of RSV F. They design several dozen dityrosine variants capable of forming intra or interprotomer dityrosine linkages, assessing these for recognition by the motavizumab antibody as well as the prefusion specific antibodies D25 and AM14. The top 16 were expressed, purified, and subjected to crosslinking, with one variant, called DT-preF, with two engineered dityrosine bonds, V185Y pairing with N428Y, and K226Y pairing with the endogenous tyrosine at Y198, appearing especially stabilized.

The formation of dityrosine crosslinks in DT-preF was partially confirmed by mass spectrometry identification of one the crosslinks, with amino acid analysis indicating the presence of a third dityrosine crosslink in each monomer, and the crosslinked protein assessed for stability at 4 C and found to be substantially more stable after 5 weeks, with immunogenicity assessment after 4 weeks of incubation showing substantial reduction in elicited neutralization titers for DS-Cav1 but not DT-preF.

DT-preF was tested versus DS-Cav1 in naïve mice, which at 1 ug/ml injection in Alum, show a 9-fold increase in relative immunogenicity. The ability of DT-preF to elicit titers in aged mice was also assessed, with increased dosing enabling similar elicited titers in aged and young mice. DT-preF was also tested in cotton rats and found to be highly potent at reducing lung titers in RSV challenged animals. Overall, the authors use dityrosine crosslinking to stabilize RSV F in its prefusion conformation and show the resultant immunogen (DT-preF) to have improved stability and immunogenicity.

It's exciting to have a new type of structure-based stabilization – enzymatically induced dityrosine crosslinks – to fix the conformation of an important vaccine antigen, and the data do suggest DT-preF is more stable and elicits improved immune responses versus DS-Cav1 in naive cotton rats and mice, the latter assessed in both standard and aged mice.

That said, several aspects of the manuscript, including referencing, described data, and order of experiments, need to be improved.

The referencing is a mess. For example, it's helpful for the authors to alert readers to the multiple prefusion stabilized RSV Fs immunogens that are in phase 3 development, such as from GSK, Pfizer, J&J and Moderna, and we are told about references 38 and 39 that (lines 83-84) that “Current RSV prefusion stabilized first generation vaccines have built upon the success of DS-Cav1 and have cleared phase 3 clinical development with licensure expected in the coming months (ref. 38,39).” However, reference 39 refers to an influenza paper (Sahni et al. Sustained Within-season Vaccine Effectiveness Against Influenza associated Hospitalization in Children: Evidence From the New Vaccine Surveillance Network, 2015-2016 Through 2019-2020. Clin Infect Dis 76, e1031-e1039, 597 (2023).) And while Ref. 38 (Papi et al.) is reasonable, the authors should cite other Phase 3 results such as from J&J (e.g. Kampmann et al. NEJM 2023), along with references to Pfizer's and Moderna's efforts. In terms of other prefusion-stabilized RSV F, “Several 2nd-generation RSV subunit vaccines are being developed that further improve upon DS-Cav1's stability (ref. 32-37).”, the authors should cite Joyce et al., Iterative structure-based improvement of a fusion-glycoprotein vaccine against RSV. Nat Struct Mol Biol. 23, 811-820 (2016). This reference describes variants of RSV F that have substantially improved stability and 4-5-fold improved immunogenicity versus DS-Cav1 and are thus likely the closest 2nd-generation version of RSV F to DT-preF. In addition to checking citations, the authors should discuss how their DT-preF compares to these 2nd generation variants.

We apologize about the discrepancies in the references. We have corrected the mislabeling of the Sahni reference and have added the suggested references as described. It is difficult for us to find an updated mRNA reference for Moderna other than press release types of reports but we have included an older reference for their mRNA program. Krarup et al describes the J and J molecule which has since been discontinued.

Critical data is not provided. Since DT-preF is the “new” immunogen that is the focus of the paper, it's critical to provide data on its development and to characterize the two dityrosine crosslinks. In terms of development, its nice to see that antigenic data on V185Y pairing with N428Y in Fig. 1B, but nothing is shown for this variant in terms of DT fluorescence (Fig. 1C).

To address the reviewer's concerns, we have updated Figure 1 to include fluorescence (Figure 1D) of the DT-preF molecule in a screening format this was done with total protein normalized supernatants as requested below.

Also, the rationale for combination is unclear; many combinations are possible, yet no mention is made of how the specific pair that makes up DT-preF was chosen.

In terms of the combination of mutations, we have added text to describe the rationale in the body text on lines 101-105 and in the results section on lines 135-137, 149-161, 164-165, and 170-171 (all described in the track changes version).

In terms of characterization, two aspects are critical, whether the crosslinks actually form and their frequency. For formation, Fig. 1C shows the fluorescence from the purposed crosslink between K226Y pairing with the endogenous tyrosine at Y198 to be one of the lower reported values, less than half that for some of the other crosslinks.

Included in the new Figure 1 using supernatants normalized for total protein expression, the fluorescence of the K226 mutant is one of the higher performing mutants.

For frequency, we can see substantial monomers and dimers on the SDS-PAGE of DT-preF (Fig. 2C). It appears from this analysis that only about half the intramolecular crosslinks form (it would be good, if the authors were to scan the gel and to report the quantification of monomer, dimer, and trimer components, and along with the frequency of intramolecular crosslink).

Tyrosines contribute significantly to Coomassie dye staining and since we are altering tyrosines, quantification by densitometry is not as accurate as we would like using SDS-PAGE. We have added text to the discussion to point out that not all bonds form in the crosslinked molecule under standard conditions (Discussion—lines 308-314).

We are told, moreover, that for the critical experiment confirming the formation of a crosslink between K226Y pairing with the endogenous tyrosine at Y198: “Amino acid analysis confirmed dityrosine’s presence in the sample and between 3 and 4 dityrosine bonds were present on average per molecule (data not shown).” Please show this data. Also, explain how this data is consistent with the SDS-PAGE analysis indicating that a substantial fraction of intermolecular crosslinks do not form.

We have incorporated the amino acid analysis data into a new figure 4A which provides the evidence directly as requested.

Overall, the authors need to show data that prompted their selection of the crosslinks in DT-preF and to define (or at least estimate with appropriate uncertainty) the frequency of their formation.

Our rationale descriptions described above should address this concern as well.

Order of experiments should be improved. Figure 4 is out of place. While I like that the authors test “shelf-life” by assessing immunogenicity from 4 C stored immunogens, I found this description prior to the characterization of relative immunogenicity in Fig. 5 to be suboptimal. Rather, it would be better to place the combined immunogenicity-stability analysis after the first description of immunogenicity. Also, it’s strange to report only the ratio of the elicited neutralizing titers, not the actual neutralizing titers themselves, please report actual neutralization titers for DT-preF and DS-Cav1 shown in Fig. 4 (for both fresh and after 4 weeks at 4 C).

We have updated the now figure 5 to include the neutralization titers as requested. The story line for the manuscript is that crosslinks impart stability and stability improves *in vivo* exposure and/or preservation of conformation that leads to the higher potency observed in the later figures. This potency also enables the overcoming of immunosenescence. This is described in the body text, results section, lines 239-241.

Therefore we have kept the order of Figures, but have now added much additional data.

Other aspects of the paper that should be improved:

1. The RSV F is a glycoprotein.

We have elaborated on this point in the body text (Results) on lines 119-121.

2. The authors utilize Cav1 mutations to help stabilize. I assume this is because the Cav1 mutations provide some prefusion stability, which is need during purification, prior to enzymatic dityrosine crosslinking. The authors should discuss whether experiments would have been successful without Cav1 and what would have happened if they introduced the dityrosine crosslinks on DS-Cav1.

Included in the new Figure 1 is data of some of the most promising mutations placed in the DS-Cav1 background. These demonstrate reduced expression levels and were not pursued. You are correct in determining that Cav1 supplied some initial stabilization which we now explicitly describe on lines 128-129.

3. In Fig. 1B and 1C, the order of the variants on the X-axis appears to be random, making it difficult to find data on specific variants tested. Please arrange in a clear order; I suggest numerical order by residue number. While it is difficult to order these mutations since they are structurally-based and do not numerically arrange nicely, we have organized them in increasing order of the first mutant listed.

4. In Fig. 1C, the authors should clarify protein quantification, and provide fluorescence as normalized by the amount of protein.

Our new Figure 1D demonstrates data from total protein normalized samples as described in the figure legend. Motavizumab (total protein) signal was used for normalization.

5. The authors should provide the fluorescence of DT-preF, as normalized by the amount of protein.

We have made this figure as requested in Supplementary Figure 1b.

6. There appear to be some typos in variant names in Fig. 1 (e.g. L18Y8 in panel C).

These have been eliminated in the new Figure 1.

7. In Fig. 5, please provide an actual number for geometric mean immunogenicity (with error) for panels A and B.

This has been done in the new Figure 6A and 6B as requested.

8. In the methods, it says 10 mice/group, but there are only 9 values shown for DT-preF in Fig. 5A/B.

The new Figure 6A also happens to have 9 mice in the most potent group. One mouse was found dead prior to the initiation of vaccinations, this is described in the statistics section on lines 588-589.

9. Since this is one of the first papers on designed dityrosine crosslinks, it would be helpful for the authors to discuss what aspect of the design correlated best with formation of crosslinks.g

Other than the presence of Tyrosine, structural proximity of the sidechains is critical for bond formation. This is described on lines 24 and 102.

10. In terms of the title, the authors report stability and potency advantages, but the safety advantage should be clarified.

Safety has been eliminated from all relevant sections.

Reviewer #3 (Remarks to the Author):

In this work, the authors use protein engineering to stabilize a viral fusion protein in its pre-fusion conformation, with the goal of developing an improved vaccine. The key outcome of this work seems to be improved storage stability of this modified protein complex. While I am not an expert on vaccine development, the authors do show results of a rat study in Figure 6, and these indicate efficacy. Results from mouse serum in Figure 5 further seem to support this, although the difference in neutralization titer between this new construct and the comparator DS-Cav1 apparently barely reaches significance ($0.01 < p < 0.05$). Still, the emphasis is on storage stability, so increased efficacy versus the comparator seems to be less critical.

Our updated potency data in Figure 6A achieves a p value of <0.01 and represents the most potent molecule described in the field as we describe in the body text on lines 316-317.

As my expertise is primarily in mass spectrometry, I focused on this aspect during my reading. Overall, this work seems interesting, and the protein MS results in Figure 3 are reasonably convincing, although I would prefer to see more data, as well as more experimental detail.

*The description of the RP-HPLC separation should be expanded. Currently, the text starting on line 357 does not even specify the HPLC instrument used (it might be the same one as for the SEC experiment, but this is not stated). The LC gradient is also not described, beyond a statement that the total analysis time was 200 minutes. MS data analysis (in particular, which software was used) is also not described in any detail.

We have expanded the description and data for the mass spectrometry analysis as requested including both methods in the body text lines 450-454 and included extra information about the methods in the Supplementary methods section.

*The text states (line 143) that 'Unique peptides were identified in the crosslinked sample, and these were selected and sequenced by tandem MS.' Figure 3, however, only shows tandem MS results for a single peptide – was this the only one that was identified? How many other unique peptides were detected that could not be identified?

As described in the discussion section, this was the only crosslinked peptide identified (lines 303-304). We have a coverage map that includes all but 8 amino acids (see attached). We would be happy to include it in the supplementary materials if you feel it is required.

*Related to the above point, the mass errors in the spectrum shown in Figure 3A seem to be quite large – I did not check all of the m/z values, but found several that seem to be off by over 100 ppm, especially at higher masses. While it does seem likely that peptide X was assigned correctly based on the spacing of the fragment signals, there do not seem to be any fragments that specifically confirm the identity of the covalently bound peptide Y (other than a possible signal of the intact peptide Y at 395.3 m/z). I appreciate that this is a targeted experiment and therefore lower instrument performance characteristics can be tolerated than in most proteomics studies. Even so, this is a sizable dataset and I would still be worried about false-positive assignments, especially if a large mass error is tolerated and if this was the one peptide (out of an unknown number of unique ones) where a plausible interpretation was found. Do the authors have additional evidence for this peptide identification?

We have included in the new Figure 3, insets which demonstrate the masses you describe above.

*Did the authors also process the peptide signals that were not unique to the crosslinked sample? If yes, what sequence coverage was obtained? I realize that it's potentially some extra work to revisit the dataset and investigate this, but it could provide insight into the overall data quality, including typical mass errors of the non-crosslinked fragments.

We have a coverage map that includes all but 8 amino acids and would be happy to include it in the supplementary materials if you feel it is required.

Here is the peptide mapping of the uncrosslinked sample.

*It would be helpful to interested readers if the MS data were made available in a repository such as PRIDE, as recommended in the editorial policies (<https://www.nature.com/nature-portfolio/editorial-policies/reporting-standards>) and the identifier provided in the 'Data availability' statement.

We do have this data set available for upload and plan to deposit it upon publication.

*Minor point: in Figure 3A, it is unclear which signal the arrow for fragment Xb8 (which I would expect at 729.4 m/z) is pointing at.

This should be corrected in the new Figure 3.

Reviewers' Comments:

Reviewer #1:

Remarks to the Author:

Engineered dityrosine-bonding of the RSV prefusion F protein imparts stability, safety, and potency advantages.

Gidwani et al. (Revised)

The manuscript has been improved in this round of revisions, especially in the biochemical characterization of the protein. However, the manuscript does not show definitive advantages of this new stabilized pre-F protein construct vs those constructs of pre-F now approved. Those advantages presented are based on moderate increase in stability at 40C, and improvement in the levels of neutralizing antibodies generated by immunization with DT-F compared with DS-CaV1 in mice (Figure 6A). Other studies in animals were non-comparative. Immune senescence study in mice (Fig. 6C-E) only indicates that higher doses of antigen are needed on elderly animals. Figure 7 shows strong lung protection after vaccination, but it is not comparative. Overall, the study presents a variation of stabilization of the F protein in the pre-fusion conformation, that increase stability and generate higher titers of neutralizing antibodies than DS-CaV1 in the mouse model. However, the study falls short to demonstrate comparative improvement in protection or safety.

Reviewer #2:

Remarks to the Author:

In "Engineered dityrosine-bonding of the RSV prefusion F protein imparts stability and potency advantages," *Gidwani et al.* describe an RSV F protein (DT-preF), stabilized by two dityrosine crosslinks, between V185Y-N428Y and Y198-K226Y.

In my prior review, I noted that the 2nd-generation constructs such as those reported in the *Joyce 2016* manuscript would be better comparison to DT-preF – and requested that "the authors should discuss how their DT-preF compares with these 2nd generation variants." However, I don't see this comparison made. In particular, it's not clear that DT-preF has stability advantages over the prior reported prefusion stabilized RSV F with multiple disulfides. That is, DT-preF appears to be a bit more stable at 4 C than DS-Cav1, while some of the 2nd generation variants with two-disulfides are much more stable than DS-Cav1.

Thus, while the initial submission trumpeted DT-preF as having stability, safety, and potency advantages, it now appears that DT-preF has potency advantages, but not safety nor stability advantages (DS-preF does not appear to be as stable as the double-disulfides molecules described by *Joyce et al.* in 2016). The title should be revised accordingly.

The potency advantage is interesting and should be discussed further. As the authors note, the elicited titers from DT-preF are second only to the DS-Cav1 nanoparticle elicited titers. Relevant to this, the authors should report the recently released interim clinical data from *Icosavax* showing only moderate titers, highlighting differences in recall versus naïve immunization. In terms of the potency advantage, this likely stems from the immunogen having multiple crosslinks. Relevant to this, it would be helpful for the authors to discuss how multiple crosslinks appears to increase elicited titers from naïve animals, as was originally noted in *Joyce et al.* in 2016 for RSV, and has now been observed in other paramyxoviruses, including human metapneumoviruses (see for example, *Li et al. 2023 PLoS Pathogens*, where a triple disulfide stabilized F elicits much higher titers than single-disulfide stabilized versions), but not with parainfluenza virus type 3, where a threshold stabilization effect was observed (*Stewart Jones et al. 2019 PNAS*).

Reviewer #3:

Remarks to the Author:

In this revised version, the authors have made a sincere effort to address my comments. My main concerns were about the reliability of the mass spectrometry analysis, and the revised Figure 3 does a lot to address this point. I would still encourage the authors to deposit the MS data into the PRIDE repository and to include the identifier in the Data Availability statement prior to publication. Apart from that, I only have a few minor points for the authors to consider.

*On line 122, there seems to be a word (I suspect 'proximity') missing in 'Since dityrosine bonds only form between tyrosine sidechains in close , we have engineered...'

*On line 172, the language is a bit terse. I suspect the authors mean 'AM14 binding' when they write 'either expressed poorly or resulted in complete loss of AM14 after crosslinking.' Adding 'binding' could make the text easier to parse for the reader.

*On lines 206-208, the authors refer to Figures 4b and 4c. There is no Figure 4c in the manuscript. Presumably, the authors mean Figure 4b bottom left and 4b right?

*On line 461, the authors write that peaks in the mass spectra were 'manually assigned using in-house software (Prottech, proprietary).' It is not obvious whether this refers to in-house software that the authors developed, or (presumably, as they are listed in the Acknowledgements) software used by Prottech Inc. Details on the software used should be included in the 'Software and code' part of the Reporting Summary (where no software is currently listed).

*On lines 481-482, I think there might be an extra 'column' in 'Samples were run on an Agilent 1260 Infinity II HPLC system using two LW-403 4D (Shodex) analytical size-exclusion columns in tandem analytical size-exclusion chromatography column at 45 °C...'

*I would encourage the authors to add the coverage map they have provided in the rebuttal letter to the Supplementary Materials, particularly as the sequence coverage is quite good. I had some trouble following the numbering though, as V185Y, Y198, K226Y, and N428Y were not in their expected positions. I suspect there is a -25 residue offset, and that the tyrosines in positions 160, 173, 201, and 403 are the critical ones. If so, it would be good if the authors could briefly address the reason for this offset in the figure caption.

We would like to thank all the reviewers for their thoughtful review of our manuscript. Their critiques have undoubtedly strengthened the manuscript significantly and we have paid careful attention to address the concerns raised. Included below is a summary of changes that we made to address the concerns. Please note that all line numbers correspond to the final submitted version of the manuscript.

Reviewer #1 (Remarks to the Author):

Engineered dityrosine-bonding of the RSV prefusion F protein imparts stability, safety, and potency advantages.
Gidwani et al. (Revised)

The manuscript has been improved in this round of revisions, especially in the biochemical characterization of the protein. However, the manuscript does not show definitive advantages of this new stabilized pre-F protein construct vs those constructs of pre-F now approved. Those advantages presented are based on moderate increase in stability at 40C, and improvement in the levels of neutralizing antibodies generated by immunization with DT-F compared with DS-CaV1 in mice (Figure 6A). Other studies in animals were non-comparative. Immune senescence study in mice (Fig. 6C-E) only indicates that higher doses of antigen are needed on elderly animals. Figure 7 shows strong lung protection after vaccination, but it is not comparative. Overall, the study presents a variation of stabilization of the F protein in the pre-fusion conformation, that increase stability and generate higher titers of neutralizing antibodies than DS-CaV1 in the mouse model. However, the study falls short to demonstrate comparative improvement in protection or safety.

We thank this reviewer for careful consideration of the manuscript. In RSV, preclinical models (mice/cotton rats) have proven very predictive of clinical outcomes lending weight to the validity of our murine potency studies. We agree that direct comparison is not made in aged mice and/or cotton rats, but strong potency data is indicated in these models as well for DT-preF.

Reviewer #2 (Remarks to the Author):

In “Engineered dityrosine-bonding of the RSV prefusion F protein imparts stability and potency advantages,”
Gidwani et al. describe an RSV F protein (DT-preF), stabilized by two dityrosine crosslinks, between V185Y-N428Y and Y198-K226Y.

In my prior review, I noted that the 2nd-generation constructs such as those reported in the Joyce 2016 manuscript would be better comparison to DT-preF – and requested that “the authors should discuss how their DT-preF compares with these 2nd generation variants.” However, I don’t see this comparison made. In particular, it’s not clear that DT-preF has stability advantages over the prior reported prefusion stabilized RSV F with multiple disulfides. That is, DT-preF appears to be a bit more stable at 4 C than DS-Cav1, while some of the 2nd generation variants with two-disulfides are much more stable than DS-Cav1.

Thus, while the initial submission trumpeted DT-preF as having stability, safety, and potency advantages, it now appears that DT-preF has potency advantages, but not safety nor stability advantages (DS-preF does not appear to be as stable as the double-disulfides molecules described by Joyce et al. in 2016). The title should be revised accordingly.

Our manuscript is the first to describe the engineering of dityrosine bonds in vaccine design. Our stability data in Supplemental figure 3A clearly demonstrates that when dityrosine bonds are introduced into the precursor molecule (preFC) making DT-preF, the melting temperature is increased in a differential scanning fluorimetry (DSF) experiment by 25 degrees Celsius to 79.53 degrees Celsius. While the DSF melting temperature of the GSK vaccine immunogen is not reported, our melting temperature exceeds the other licensed RSV vaccine immunogen (Abrysvo, Pfizer, Tm=69.2 degrees Celsius). While there is no reported DSF melting temperature

for the DS-2 molecules described in Joyce et al. and referenced by the reviewer, our current title, "Engineered dityrosine-bonding of the RSV prefusion F protein imparts stability and potency advantages", simply points out that stability is increased through dityrosine bond introduction and does not boast to be the most stable molecule in the field. Therefore, we feel the title is accurate as described and do not wish to change it further.

The potency advantage is interesting and should be discussed further. As the authors note, the elicited titers from DT-preF are second only to the DS-Cav1 nanoparticle elicited titers. Relevant to this, the authors should report the recently released interim clinical data from Icosavax showing only moderate titers, highlighting differences in recall versus naïve immunization. In terms of the potency advantage, this likely stems from the immunogen having multiple crosslinks. Relevant to this, it would be helpful for the authors to discuss how multiple crosslinks appears to increase elicited titers from naïve animals, as was originally noted in Joyce et al. in 2016 for RSV, and has now been observed in other paramyxoviruses, including human metapneumoviruses (see for example, Li et al. 2023 PLoS Pathogens, where a triple disulfide stabilized F elicits much higher titers than single-disulfide stabilized versions), but not with parainfluenza virus type 3, where a threshold stabilization effect was observed (Stewart Jones et al. 2019 PNAS).

Our titers exceed those of the DS-Cav1 nanoparticle (DT-preF= 11X, RSV nanoparticle= 9X) in murine studies. However, based on changes made to the DS-Cav1 base molecule currently in the Icosavax clinical trial, we cannot attribute the poorer clinical performance directly to differences in naïve vs recall immunizations. In fact, it seems more likely that the changes made to the molecule have reduced overall potency compared to the initially reported molecule in Marcandalli et al.. Therefore, we will not comment further on this topic. At the reviewer's request, we have added text (lines 310 to 322) to address the reviewer's suggestion to delve into the topic of additional crosslinks improving potency based on the 3 papers referenced by the reviewer. (Joyce, Li (Ou), and Stewart Jones). We hope this addresses the reviewer's suggestion sufficiently.

Reviewer #3 (Remarks to the Author):

In this revised version, the authors have made a sincere effort to address my comments. My main concerns were about the reliability of the mass spectrometry analysis, and the revised Figure 3 does a lot to address this point. I would still encourage the authors to deposit the MS data into the PRIDE repository and to include the identifier in the Data Availability statement prior to publication. Apart from that, I only have a few minor points for the authors to consider.

At the reviewer's request, we have uploaded the data set for uncrosslinked (preF^C) and crosslinked (DT-preF) RSV F proteins into the PRIDE database (accession number: PXD048897. We have the following reviewer access credentials: **Username:** reviewer_pxd048897@ebi.ac.uk **Password:** ghck6FyU.

We have updated the data availability statement to reflect our submission to the PRIDE database. We have also added text in the methods section of the manuscript, lines 448-451 to reference the deposition and accession number.

*On line 122, there seems to be a word (I suspect 'proximity') missing in 'Since dityrosine bonds only form between tyrosine sidechains in close , we have engineered...'

We thank the reviewer for their careful attention to detail and have made the appropriate change.

*On line 172, the language is a bit terse. I suspect the authors mean 'AM14 binding' when they write 'either expressed poorly or resulted in complete loss of AM14 after crosslinking.' Adding 'binding' could make the text easier to parse for the reader.

We thank the reviewer for their careful attention to detail and have made the appropriate change.

*On lines 206-208, the authors refer to Figures 4b and 4c. There is no Figure 4c in the manuscript. Presumably, the authors mean Figure 4b bottom left and 4b right?

We thank the reviewer for their careful attention to detail and have made the appropriate change.

*On line 461, the authors write that peaks in the mass spectra were 'manually assigned using in-house software (Prottech, proprietary).' It is not obvious whether this refers to in-house software that the authors developed, or (presumably, as they are listed in the Acknowledgements) software used by Prottech Inc. Details

on the software used should be included in the 'Software and code' part of the Reporting Summary (where no software is currently listed).

The mass spectrometry data was contracted to Prottech Inc. and they used their own, proprietary, in-house software to analyze and assign the mass spectra. We have now made this clear in the software and code section. We hope this satisfies the request. Calder did not generate any mass spectrometry analysis software.

*On lines 481-482, I think there might be an extra 'column' in 'Samples were run on an Agilent 1260 Infinity II HPLC system using two LW-403 4D (Shodex) analytical size-exclusion columns in tandem analytical size-exclusion chromatography column at 45 °C...'

We thank the reviewer for their careful attention to detail and have made the appropriate change.

*I would encourage the authors to add the coverage map they have provided in the rebuttal letter to the Supplementary Materials, particularly as the sequence coverage is quite good. I had some trouble following the numbering though, as V185Y, Y198, K226Y, and N428Y were not in their expected positions. I suspect there is a -25 residue offset, and that the tyrosines in positions 160, 173, 201, and 403 are the critical ones. If so, it would be good if the authors could briefly address the reason for this offset in the figure caption.

We have included the coverage map in the Supplemental data section as requested (Supplemental figure 4). In the figure legend, we describe that the mapped sequence is for the mature protein and lacks the Signal peptide and the p27 proteolytically-processed piece. In the listed sequence, the mutated tyrosines are at positions, (Y160 pairs with Y403 and Y173 pairs with Y201) in the abridged/mature sequence. We put a box (mutated residue) or a circle (endogenous) around these to clarify.